# Report

# Reevaluating the senolytic activity of a GLS1 inhibitor and an anti-PD-1 antibody: toward greater reproducibility and methodological rigor

Shimpei Kawamoto [ID][1,2,3✉], Haruki Horiguchi [ID][4], Daisuke Torigoe [ID][5], Masahiro Wakita[1], Koyu Ito[1], Sho Sugawara [ID][6], Xiangyu Zhou[6], Takumi Mikawa [ID][7], Jeong Hoon Park[1], Birte Kristin Jung[1], Yumiko Okumura[1], Hideka Miyagawa[1], Mikako Maruya[8], Nozomi Hori[1], Ken Uemura[1], Masataka Sugimoto [ID][9], Michiyuki Matsuda [ID][10], Naoki Mochizuki [ID][11], Hiroshi Kondoh [ID][7], Akiko Takahashi [ID][6], Yuichi Oike[4] & Eiji Hara [ID][1,12✉]

## Abstract

**The discovery of the senescence-associated secretory phenotype (SASP) has reshaped our understanding of cellular senescence, shifting its role from a solely tumor-suppressive mechanism to a potential driver of chronic inflammation and age-related diseases. Accordingly, senolytic drugs, which selectively eliminate senescent cells, have garnered considerable interest due to promising pre-clinical studies. However, concerns remain regarding the reproducibility and generalizability of these findings. In this cross-laboratory study, we rigorously tested the senolytic efficacy of a GLS1 inhibitor and an anti-PD-1 antibody—agents previously reported to reduce the burden of p16[INK4a]-positive senescent cells and improve health outcomes in aged mice. Contrary to earlier reports, our study demonstrates that neither GLS1 inhibition nor PD-1 blockade significantly reduced p16[INK4a]-positive cell burden or improved aging-related health parameters. Although we do not seek to discredit prior work, our results underscore the need for rigorous design, standardized protocols, and independent validation to ensure reliable senolytics before clinical translation.**

**Keywords** Cellular Senescence; Senolysis; Senolytic Drug; Aging; Reproducibility
**Subject Categories** Autophagy & Cell Death; Molecular Biology of Disease; Pharmacology & Drug Discovery

See also: Y Johmura et al

## Introduction

Cellular senescence is a state of stable cell-cycle arrest that can be triggered by various potentially oncogenic stimuli, including telomere erosion, radiation, oxidative stress, and oncogene activation (Campisi and d'Adda di Fagagna, 2007; Gorgoulis et al, 2019; Serrano and Blasco, 2001). Furthermore, many key genes involved in the induction of cellular senescence are classified as tumor suppressor genes, suggesting that cellular senescence plays a crucial role in tumor suppression (Collado and Serrano, 2010; He and Sharpless, 2017). Unlike apoptosis, however, senescent cells do not undergo immediate cell death; consequently, they gradually accumulate in the body with age (Krishnamurthy et al, 2004; Yamakoshi et al, 2009). Importantly, senescent cells are not simply arrested in the cell-cycle but also exhibit the senescence-associated secretory phenotype (SASP), characterized by the secretion of inflammatory cytokines, chemokines, growth factors, and extracellular matrix-degrading enzymes (Acosta et al, 2008; Coppe et al, 2008; Kuilman et al, 2008). Thus, excessive senescent cell accumulation can drive chronic inflammation in surrounding tissues through SASP, potentially contributing to functional decline and promoting the onset of various age-related diseases, including cancer (Chan and Narita, 2019; Lee and Schmitt, 2019; Watanabe et al, 2017). Indeed, van Deursen and colleagues developed a mouse model harboring a transgene in which an apoptosis-inducing gene is driven by the 2.7 kb promoter of *p16[INK4a]*, a key senescence-inducing gene (Baker et al, 2011). Using this model, they genetically demonstrated that the ablation of *p16[INK4a]*-expressing cells reduced the incidence of aging-associated cancers and extended health span (Baker et al, 2016). Since this report, the concept that eliminating senescent cells can extend health span has garnered significant attention (Naylor et al, 2013). Consequently, the development of

[1]Research Institute for Microbial Diseases, The University of Osaka, Suita, Japan. [2]Institute of Development, Aging and Cancer, Tohoku University, Sendai, Japan. [3]Organization for Advanced Studies, Tohoku University, Sendai, Japan. [4]Graduate School of Medical Sciences, Kumamoto University, Kumamoto, Japan. [5]Institute of Resource Development and Analysis, Kumamoto University, Kumamoto, Japan. [6]Cancer Institute, Japanese Foundation for Cancer Research, Tokyo, Japan. [7]Kyoto University Hospital, Kyoto, Japan. [8]RIKEN Center for Integrative Medical Sciences, Yokohama, Japan. [9]National Center for Geriatrics and Gerontology, Obu, Japan. [10]Graduate School of Medicine, Kyoto University, Kyoto, Japan. [11]Research Institute, National Cerebral and Cardiovascular Center, Suita, Japan. [12]Immunology Frontier Research Center, The University of Osaka, Suita, Japan.
✉E-mail: shimpei.kawamoto@tohoku.ac.jp; ehara@biken.osaka-u.ac.jp

senolytic drugs, which selectively eliminate senescent cells, has been actively pursued worldwide (Chaib et al, 2022; Wang et al, 2022; van Deursen, 2019). However, senescent cells exhibit considerable heterogeneity (Tao et al, 2024; Wechter et al, 2023), and some contribute beneficially to specific physiological processes, such as promoting wound healing, maintaining blood-tissue barriers, and activating immune responses, depending on the biological context (Grosse et al, 2020; Kang et al, 2011; Krizhanovsky et al, 2008; Reyes et al, 2022; Zhao et al, 2024). Therefore, the indiscriminate elimination of senescent cells may result in severe adverse effects, underscoring the need for a more selective and cautious approach in the development of senolytic drugs (Amor et al, 2020; Suda et al, 2021; Yoshida et al, 2020).

To date, more than 20 senolytic drug candidates have been reported, with some shown to improve health in aged mice (Chaib et al, 2022; Wang et al, 2022; Power et al, 2023). For example, BPTES, a GLS1 inhibitor, almost completely eliminated senescent cells in culture at a 10 μM concentration (Johmura et al, 2021). Furthermore, BPTES administration in aged mice reportedly reduced the burden of $p16^{INK4a}$-expressing cells across multiple organs, resulting in significant improvements in health parameters (Johmura et al, 2021). However, our analysis comparing the activities and specificities of several senolytic drugs suggested that BPTES exhibits relatively low specificity and limited senolytic activity (Wakita et al, 2026). In biological experiments, subtle differences in experimental conditions and technique proficiency may influence the quality of the data. Therefore, we deemed it essential to evaluate these data in independent laboratories.

## Results and discussion

To validate the senolytic effects of BPTES, which appears to be a highly potent senolytic agent (Johmura et al, 2021), we conducted independent replication studies in the Hara (Osaka), Takahashi (Tokyo), and Kondoh (Kyoto) laboratories. Experiments conducted in the Hara and Takahashi laboratories using multiple cell types—including IMR-90 cells, a normal human diploid fibroblast (HDF) line also used by Johmura et al (2021)—revealed that treatment with BPTES (10 μM) resulted in cell death in approximately 50% of IMR-90 senescent cells and 30% of TIG-3 senescent cells, regardless of the method used to induce senescence (Fig. 1A–D). However, at this concentration, BPTES markedly inhibited the proliferation of non-senescent (control) cells and induced a certain degree of cell death (Figs. 1B,D and EV1). In Kondoh's laboratory, similar results were observed in senescent TIG-1 cells—another type of HDF—induced by the DNA-damaging agent etoposide (Fig. 1E,F). Notably, increasing the concentration of BPTES to 20 μM led to a slight increase in senescent cell death. However, at this concentration, the proliferation of control cells was also more severely inhibited than at 10 μM (Fig. 1B,D). Conversely, although 1 μM BPTES had no growth-inhibitory effect on control cells, it also failed to induce cell death in senescent cells (Fig. 1B,D). Furthermore, given that a GLS1 inhibitor reportedly induced senescent cell death by impairing the neutralization of intracellular acidosis, we next examined whether an incubation in alkaline medium (pH 8.5), as described by Johmura et al (2021), could attenuate BPTES-induced senescent cell death. Contrary to their report (see Fig. 1G,H in Johmura et al, 2021), the cell death-

inducing effect of 10 μM BPTES was completely unaffected by the incubation in alkaline medium (Fig. 1G). This result was consistently reproduced in independent experiments conducted in both Hara's and Takahashi's laboratories. Since differences between BPTES lots might have influenced the results, we tested six different lots from two independent suppliers. Nevertheless, our findings remained largely consistent across all batches (Figs. EV1 and EV2). Collectively, these results suggest that the GLS1 inhibitor BPTES does not exhibit truly selective toxicity toward senescent cells, prompting us to reevaluate the previous report claiming that BPTES administration reduces the $p16^{INK4a}$-expressing cell population and ameliorates aging-related phenotypes in aged mice (Johmura et al, 2021).

To address this concern, we attempted to replicate the experiment previously reported by Johmura et al (2021), in which BPTES (0.25 mg per 20 g body weight in 200 μl) or vehicle control (200 μl of 10% DMSO in corn oil) was administered to 80-week-old male C57BL/6 mice three times per week for one month, in Hara's laboratory (Fig. 2A,B). However, in contrast to the previous report (see Fig. S13A–C in Johmura et al, 2021), BPTES administration did not significantly reduce $p16^{INK4a}$ expression in the liver, lung, or kidney, although the levels of $p16^{INK4a}$ expression increased in these tissues with age (Fig. 2C). Furthermore, although the same group reported that BPTES administration to aged mice resulted in a visibly more youthful appearance, we were unable to confirm such changes in our study (Fig. 2D). Because aged mice exhibit high inter-individual variability, the resulting data dispersion poses a considerable risk of arbitrary or erroneous interpretation. To minimize potential bias, we designed experiments in which the analysts were blinded to the treatment allocation (BPTES or vehicle control). Specifically, in Oike's laboratory (Kumamoto), 80-week-old male C57BL/6 mice were administered BPTES (0.25 mg per 20 g body weight in 200 μl) or vehicle control (200 μl of 10% DMSO in corn oil) three times per week for one month. Following euthanasia, organs were harvested, snap-frozen, and shipped to Hara's laboratory with coded identifiers to maintain blinding. In Hara's laboratory, RNA was extracted from these tissues and subjected to RT-qPCR analysis. All data were then transmitted to Oike's laboratory, where the treatment codes were unblinded, and statistical analyses were performed (Fig. 3A–C). Consistent with our previous observations (Fig. 2), no significant differences in $p16^{INK4a}$ expression in the liver, lung, or kidney tissues, or in external appearance, were observed between BPTES-treated and vehicle-treated mice (Fig. 3C,D). Furthermore, although BPTES treatment slightly reduced body weight, no significant differences were observed in physical parameters, such as grip strength, between the treatment groups (Fig. 3B,E). At present, the reasons underlying the discrepancy between our results and those reported by Johmura et al (2021) remain unclear. While our findings do not necessarily invalidate the data presented by Johmura et al (2021), they underscore the need for cautious interpretation of the senolytic effects of GLS1 inhibitors.

Given that the same group previously reported a reduction in senescent cells following anti-PD-1 antibody treatment in aged mice (Wang et al, 2022), we next sought to assess the reproducibility of this finding as well. To this end, 18-month-old male C57BL/6 mice were intraperitoneally injected with either isotype control IgG or anti-mouse PD-1 antibody (Chamoto et al, 2023) (250 μg per dose, 8 injections over a 3-week period) in Hara's

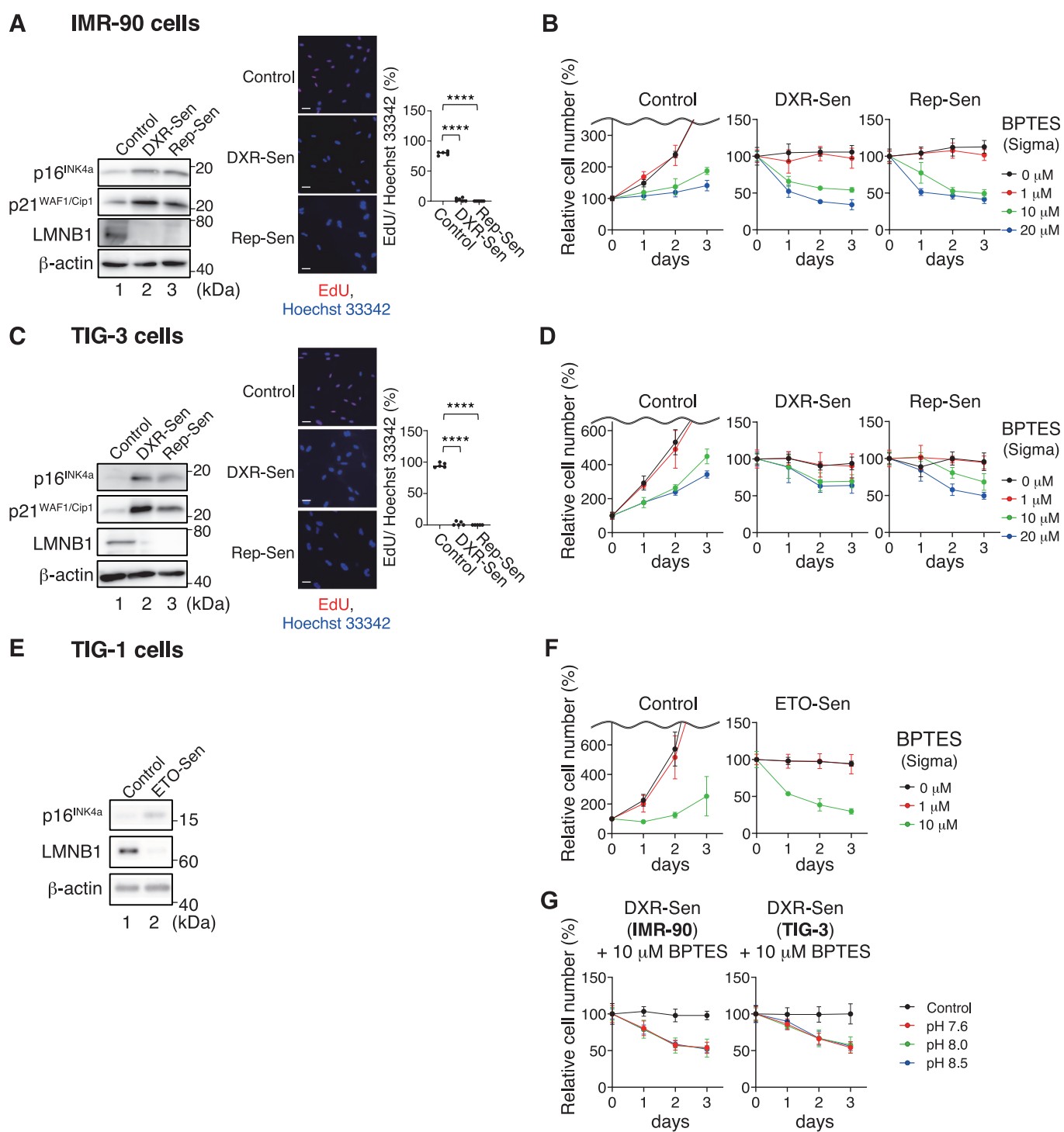

laboratory (Fig. 4A). One day after the final injection, the mice were euthanized, and $p16^{INK4a}$ expression in the liver, lung, and kidney was quantified by RT-qPCR. However, in contrast to the previous report by Wang et al (see Fig. 4a, Extended Data Fig. 7c, d in 2022), the administration of the anti-mouse PD-1 antibody did not result in a statistically significant reduction in the levels of $p16^{INK4a}$ gene expression in the liver, lung, or kidney (Fig. 4B). Furthermore, although we also examined $p16^{INK4a}$ expression in aged (24-month-old) PD-1-deficient mice (Nishimura et al, 2001), no statistically significant differences were observed in the liver or kidney compared to age-matched wild-type controls (Fig. 4C,D). Interestingly, however, the $p16^{INK4a}$ levels were unexpectedly increased, rather than decreased, in the lungs of PD-1-deficient mice (Fig. 4D). To ensure the validity of these results, we also designed blinded experiments in which the analysts were unaware of the treatment allocation (anti-PD-1 antibody or IgG control). Specifically, in

**Figure 1.  BPTES reduces proliferation and survival of cultured cells.**

(A–G) Early-passage (Control) IMR-90, TIG-3, and TIG-1 cells were rendered senescent by serial passaging (Rep-Sen), treatment with 250 ng/ml doxorubicin for 10 days (DXR-Sen), or treatment with 100 µM etoposide for 2 days followed by 10 days of culture in normal medium (ETO-Sen). These cells were subjected to western blot analysis using the indicated antibodies (A, C, E) and EdU incorporation assay (A, C) to confirm their senescent state. β-actin was used as a loading control (A, C, E). Quantification of EdU-positive cells is shown in the right panels (A, C). Control and senescent cells were treated with BPTES at the indicated concentrations for 3 days, and cell numbers were monitored throughout the experimental period (B, D, F). DXR-Sen IMR-90 and TIG-3 cells were treated with 10 µM BPTES under different pH conditions, and the number of surviving cells was counted. Control indicates senescent cells without BPTES treatment (G). Data are presented as mean ± s.d. (A, C: $n = 5$–6; B, D, G: $n = 4$; F: $n = 3$). All of the experiments were repeated at least twice, independently, with similar results. Statistical significance was determined by one-way ANOVA with Dunnett's test (A, C). Control vs. DXR-Sen, $****p < 0.0001$; Control vs. Rep-Sen, $****p < 0.0001$ (A, C). Scale bars, 10 µm (C, E). Source data are available online for this figure.

Oike's laboratory, 18-month-old male C57BL/6 mice were intra-peritoneally injected with either isotype control IgG or anti-mouse PD-1 antibody (250 µg per dose, 8 injections over a 3-week period) (Fig. 5A,B). After euthanasia, liver and lung tissues were harvested, immediately snap-frozen, and shipped to Hara's laboratory with coded identifiers to maintain blinding. In Hara's laboratory, these tissues were subjected to RT-qPCR or immunohistochemical analysis using an anti-p16$^{INK4a}$ antibody. For the immunohisto-chemical analysis, image data were analyzed using computational methods in Matsuda's laboratory (Kyoto). All data were then transmitted to Oike's laboratory, where the treatment codes were unblinded, and statistical analyses were conducted. Consistent with our previous observations (Fig. 4), we detected no statistically significant differences in p16$^{INK4a}$ RNA or protein levels in liver and lung tissues between the anti-PD-1 antibody-treated and control IgG-treated mice (Fig. 5C–E). Furthermore, although the anti-PD-1 antibody treatment slightly reduced the body weight, no significant differences were observed between the treatment groups in external appearance or physical parameters, such as grip strength (Fig. 5B,F,G). This is broadly consistent with a recent report indicating that anti-PD-1 antibody administration failed to reduce the number of p16$^{INK4a}$-positive cells in inflamed lung tissue in mice (Majewska et al, 2024). Together, although our findings do not entirely refute those reported by Wang et al (2022), they highlight the need for cautious interpretation of the senolytic effects of anti-PD-1 antibodies.

At present, the reasons underlying the discrepancy between our results and those reported by Johmura et al (2021) and Wang et al (2022) remain unclear. It is well recognized that biological experiments are often difficult to reproduce, as even subtle differences in experimental conditions can alter outcomes. In this study, however, the BPTES treatment experiments in aged mice were conducted under the guidance of Dr. Sugimoto, who was directly responsible for the corresponding experiments in Johmura et al (2021) and is also a co-author of the present study. We therefore consider it unlikely that methodological discrepancies played a major role in this case. By contrast, aged mice are known to exhibit pronounced inter-individual variability (Hamieh et al, 2021), which may have contributed to the differences between the data of Johmura et al (2021) and Wang et al (2022) and our own. Senolytic drugs, however, are ultimately intended for use in humans, who display even greater inter-individual variability than laboratory mice. Therefore, experimental designs should aim to yield robust results that minimize the influence of both inter-individual variability and experimenter-dependent variability. In this regard, cross-laboratory, blinded studies, such as the one conducted here, represent a promising strategy.

Another consideration is age. Eighty-week-old and 18-month-old 57BL/6 mice cannot be regarded as genuinely old. We nevertheless selected this age to ensure comparability with Johmura et al (2021) and Wang et al (2022). Although Johmura et al (2021) also examined 100-week-old mice, their analyses were limited to adipose tissue, where abundant macrophages complicate inter-pretation due to p16$^{INK4a}$ expression and SA-β-gal activity independent of senescence (Hall et al, 2017). To avoid these confounders, we focused on lungs, liver, and kidney in 80-week-old mice. Consistent with Johmura et al (2021), we observed increased p16$^{INK4a}$ expression in these tissues with age (Fig. 2C), although no reduction was observed after BPTES treatment.

An important point to note is that Wang et al (2022) did not directly assess endogenous p16$^{INK4a}$ expression, but instead quantified tdTomato reporter–positive cells using the p16-CreERT2-tdTomato mouse model (Omori et al, 2020). Notably, this system is designed for lineage tracing rather than real-time monitoring of p16$^{INK4a}$ expression. Therefore, tdTomato positivity reflects cells that have previously activated the p16$^{INK4a}$ promoter, and may not accurately represent the current endogenous p16$^{INK4a}$ expression levels. To investigate this possibility, we reanalyzed the single-cell RNA sequencing (scRNA-seq) data from the livers and kidneys of p16-CreERT2-tdTomato mice previously reported by Omori et al (2020). However, we could not detect any expression of the Cdkn2a gene, which encodes p16$^{INK4a}$. This raises uncertainty as to whether tdTomato expression reliably reflects endogenous p16$^{INK4a}$ expression. We therefore emphasize the importance of directly assessing the endogenous p16$^{INK4a}$ expression to ensure the accurate interpretation of experimental results. Furthermore, p16$^{INK4a}$ expression alone cannot be regarded as a definitive marker of aging in mice (Hall et al, 2017). Nevertheless, our study was not designed to establish such a marker, but rather to test the reproducibility of the key findings reported by Johmura et al (2021) and Wang et al (2022), which suggested the strong senolytic activities of BPTES and anti-PD1 antibody—claims that have attracted substantial public attention and hold profound societal consequences. Given this context, we deliberately focused on reproducing the datasets most central to those reports. In both studies, p16$^{INK4a}$ expression was employed as a key marker of cellular senescence, while grip strength served as an indicator of organismal aging (Johmura et al, 2021; Wang et al, 2022). Moreover, the same groups have emphasized through mass media that treatments with these drugs induce an apparent rejuvenation in the physical appearance of mice. Because such photographic evidence can strongly influence public perception, we have included photographs of all mice before and after treatment with either BPTES or anti-PD1 antibody (Figs. 2D, 3D, and 5G).

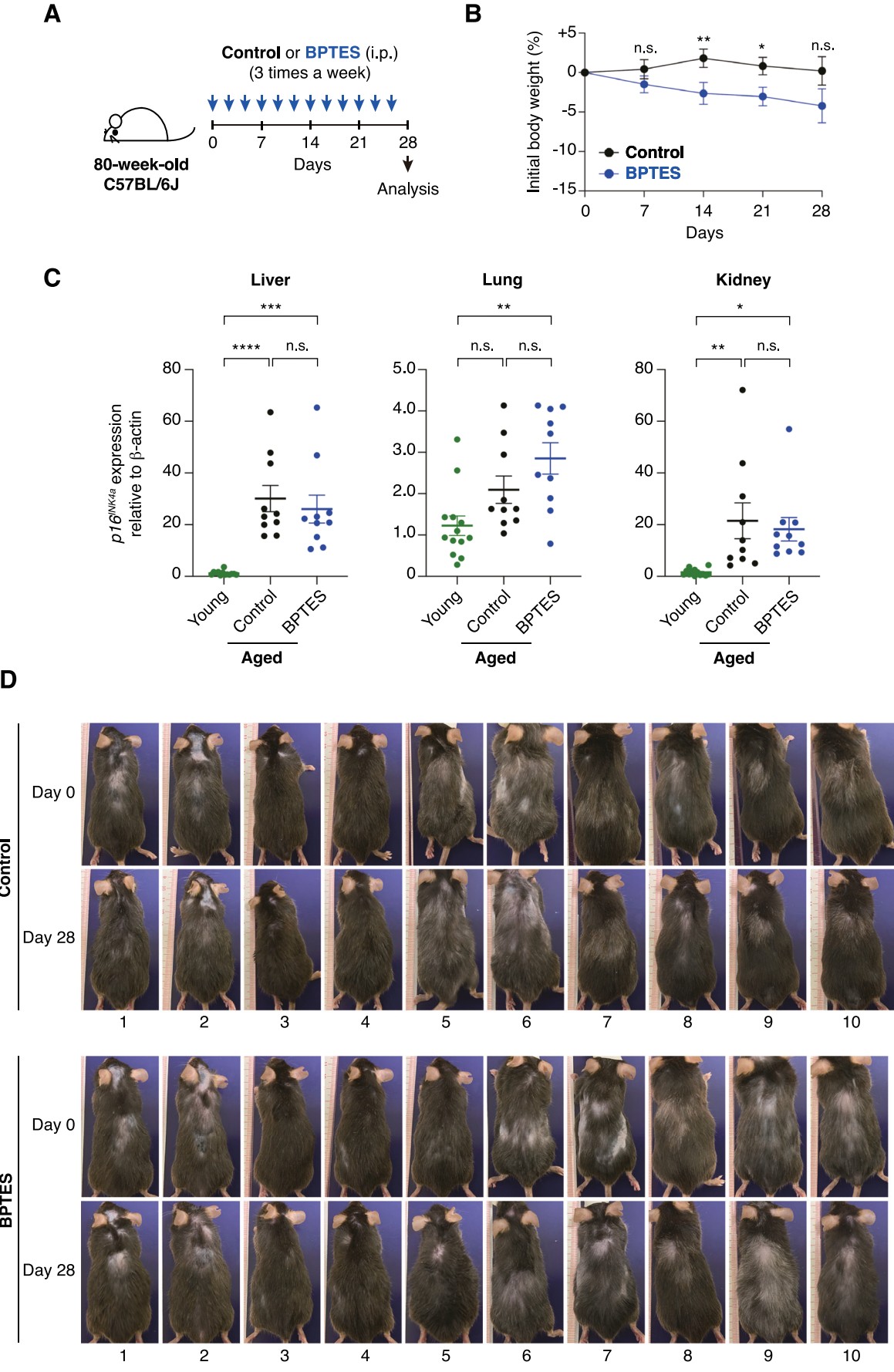

**A** Control or BPTES (i.p.) (3 times a week)

80-week-old C57BL/6J

Days 0 7 14 21 28

Analysis

**B** Initial body weight (%)

Days

● Control
● BPTES

**C**

Liver

$p16^{INK4a}$ expression relative to β-actin

Young Control BPTES
Aged

Lung

Young Control BPTES
Aged

Kidney

Young Control BPTES
Aged

**D**

Control

Day 0

Day 28

1 2 3 4 5 6 7 8 9 10

BPTES

Day 0

Day 28

1 2 3 4 5 6 7 8 9 10

**Figure 2. Administration of BPTES fails to reduce p16INK4a expression in aged mice.**

(A) Timeline of the experimental procedure for BPTES administration in 80-week-old C57BL/6 J mice. (B) Body weight changes over time, shown relative to the starting weight, in mice treated with BPTES or vehicle control (control, $n = 10$; BPTES, $n = 10$). (C) Expression of $p16^{INK4a}$ in the liver, lung, and kidney of young (8-week-old) and aged (80-week-old) mice treated with BPTES or vehicle control, as assessed by RT-qPCR (young, $n = 13$; control, $n = 10$; BPTES, $n = 10$). Fold changes were calculated using the ΔΔCT method relative to the young group. (D) Representative images of all mice taken before (day 0) and after (day 28) the completion of BPTES or vehicle administration (control, $n = 10$; BPTES, $n = 10$). Data are presented as mean ± s.e.m. Statistical significance was determined by the Mann–Whitney $U$ test (B) and one-way ANOVA followed by Šídák's multiple comparisons test (C). Day7, $p = 0.36$; Day14, **$p = 0.0096$; Day21, *$p = 0.030$; Day28, $p = 0.066$ (B). Young vs. Control: ****$p < 0.0001$ (Liver), $p = 0.15$ (Lung), **$p = 0.0074$ (Kidney); Young vs. BPTES: ***$p = 0.0002$ (Liver), **$p = 0.0021$ (Lung), *$p = 0.029$ (Kidney); Control vs. BPTES: $p = 0.87$ (Liver), $p = 0.29$ (Lung), $p = 0.94$ (Kidney) (C). n.s., not significant ($p \geq 0.05$). Source data are available online for this figure.

Why is it challenging to confirm the reproducibility of experimental data in senolytic research (Harrison et al, 2024; Kovacovicova et al, 2018)? One possible explanation is that inter-individual variability in mice increases with age (Hamieh et al, 2021), in parallel with the heterogeneity of senescent cells (Tao et al, 2024; Wechter et al, 2023). To address this, we recommend increasing the number of animals to account for such variability and conducting blinded experiments in collaboration with multiple laboratories to minimize researcher bias (i.e., ensuring that experimenters are unaware of treatment allocation). Finally, we emphasize that our intention is not to discredit previous work by any specific group, but rather to provide an independent and rigorous validation, which we believe is valuable for the field as senolytic therapies advance toward clinical applications.

## Methods

### Reagents and tools table

| Reagent/Resource | Reference or Source | Identifier or Catalog Number |
| --- | --- | --- |
| **Experimental models** | | |
| TIG-1 | JCRB Cell Bank | JCRB0501 |
| TIG-3 | JCRB Cell Bank | JCRB0506 |
| IMR-90 | JCRB Cell Bank | JCRB9054 |
| C57BL/6 J (*M. musculus*) | CLEA Japan, the National BioResource Project of the Ministry of Education, Culture, Sports, Science and Technology in Japan | C57BL/6JJcl |
| C57BL/6 N (*M. musculus*) | CLEA Japan | C57BL/6NJcl |
| *PD-1*-deficient mice (*M. musculus*) | Nishimura et al, 2001 | |
| *p16/p21* double knockout mice (*M. musculus*) | Kawamoto et al, 2023 | |
| **Antibodies** | | |
| Mouse anti-β-actin | Sigma-Aldrich | A5316 |
| Mouse anti-β-actin | Sigma-Aldrich | A4700 |
| Rabbit anti-Lamin B1 | Abcam | ab16048 |
| Mouse anti-p16INK4a | Santa Cruz | sc-56330 |
| Mouse anti-p16INK4a | IBL | 11104 |
| Rabbit anti-p21Waf1/Cip1 | Cell Signaling Technology | 2947 |

| Reagent/Resource | Reference or Source | Identifier or Catalog Number |
| --- | --- | --- |
| HRP-conjugated anti-rabbit or mouse IgG | Cell Signaling Technology | 7074, 7076 |
| GoInVivo™ Purified Rat IgG2a, κ Isotype Ctrl Antibody | BioLegend | 400563 |
| GoInVivo™ Purified anti-mouse CD279 (PD-1) Antibody | BioLegend | 135234 |
| Rabbit anti-p16INK4a | Abcam | ab211542 |
| **Oligonucleotides and other sequence-based reagents** | | |
| Mouse *Actb* primers | Kawamoto et al, 2023 | Methods and Protocols |
| Mouse *p16INK4a* primers | Kawamoto et al, 2023 | Methods and Protocols |
| **Chemicals, Enzymes and other reagents** | | |
| DMEM | Nacalai tesque | 08458-16 |
| MEM | Nacalai tesque | 21442-25 |
| Fetal bovine serum | MP Biomedicals | 2917345H |
| Fetal bovine serum | Sigma-Aldrich | 173012 |
| MEM Nonessential Amino Acid Solution | Sigma-Aldrich | 06344-56 |
| Penicillin-streptomycin-glutamine | Sigma-Aldrich | P4333 |
| Penicillin-streptomycin-glutamine | Gibco | C10378-016 |
| Doxorubicin | FUJIFILM Wako Pure Chemical Corporation | 046-21523 |
| Etoposide | Sigma-Aldrich | 341205 |
| BPTES | Sigma-Aldrich | SML0601 |
| BPTES | Cayman | 19284 |
| Protease inhibitor cocktail | Nacalai Tesque | 25955-11 |
| Protein Quantification Assay | Takara Bio | 740967.250 |
| PVDF membrane | EMD Millipore | IPVH00010 |
| Amersham ECL Prime or Select reagent | GE Healthcare | RPN2236 or RPN2235 |
| Alexa Fluor 488 Click-iT EdU Imaging Kit | Thermo Fisher Scientific | C10637 |
| DAPI | Dojindo | 340-07971 |
| FITC-conjugated annexin V | Thermo Fisher Scientific | A13199 |

| Reagent/Resource | Reference or Source | Identifier or Catalog Number |
|---|---|---|
| Bouin's fixative | Muto Pure Chemicals | 33142 |
| PathoClean | FUJIFILM Wako Pure Chemical Corporation | 161-28321 |
| ImmPRESS HRP horse anti-rabbit IgG polymer detection kit | Vector Laboratories | MP-7401 |
| ImmPACT DAB substrate kit | Vector Laboratories | SK-4105 |
| Hematoxylin | Muto Pure Chemicals | 30002 |
| PARAmount | FALMA | 308-400 |
| RNeasy Mini Kit | QIAGEN | 74106 |
| PrimeScript RT Reagent Kit with gDNA Eraser | Takara Bio | RR047A |
| TB Green Premix Ex Taq II | Takara Bio | RR820A |
| **Software** | | |
| Fiji | Schindelin et al, 2012 | |
| MATLAB R2024a | The MathWorks, Inc. | |
| Cellpose | Stringer et al, 2021 | |
| Prism v10.4.0 | GraphPad software | |
| **Other** | | |
| All-in-One Fluorescence Microscope | Keyence | BZ-X710 |
| Amersham ImageQuant 800 system | Cytiva | |
| Grip strength meters | Columbus Instruments | |
| Microtome | Leica Biosystems | |
| SLIDEVIEW VS200 Universal Whole Slide Imaging Scanner | Evident | |
| Thermal Cycler Dice Real Time System III | Takara Bio | |

## Cell culture

Normal human diploid fibroblasts (HDFs), including TIG-1 (JCRB0501), TIG-3 (JCRB0506), and IMR-90 (JCRB9054), were obtained from the Japanese Collection of Research Bioresources (JCRB) Cell Bank. TIG-3 and IMR-90 cells were cultured in Dulbecco's modified Eagle's medium (DMEM; 08458-16, Nacalai Tesque) supplemented with 10% fetal bovine serum (FBS; 2917345H, MP Biomedicals, or 173012, Sigma-Aldrich) and 100 U/ml penicillin-streptomycin (P4333; Sigma-Aldrich). TIG-1 cells were cultured in Modified Eagle's Medium (MEM; 21442-25, Nacalai Tesque) supplemented with 10% FBS, MEM Nonessential Amino Acid Solution (06344-56, Sigma-Aldrich), and penicillin-streptomycin-glutamine (P4333, Sigma-Aldrich, or C10378-016, Gibco). Cellular senescence was induced by treating cells with 250 ng/ml doxorubicin (046-21523, FUJIFILM Wako Pure Chemical Corporation) for 10 days, or 100 μM etoposide (341205, Sigma-Aldrich) for 2 days followed by a 10-day recovery period in normal culture medium. The pH of the culture medium was adjusted as

needed using NaOH or HCl. To evaluate the senolytic activity of BPTES (SML0601, Sigma-Aldrich, or 19284, Cayman), $1.5 \times 10^5$ cells were seeded in 12-well plates with grids, and cell numbers at fixed positions were recorded daily using a BZ-X710 All-in-One Fluorescence Microscope (Keyence). Relative cell number was calculated as follows: the average cell count on day 0 was defined as 100%, and the number of cells at each time point was expressed as a percentage relative to the day 0 value.

## Western blotting analysis

Cells were lysed in RIPA buffer supplemented with 1% protease inhibitor cocktail (25955-11, Nacalai Tesque). Protein concentrations were determined using the Protein Quantification Assay (740967.250, Takara Bio). Samples were denatured in Laemmli sample buffer by heating at 95 °C for 5 min. Proteins were separated by SDS–polyacrylamide gel electrophoresis and transferred onto PVDF membranes (IPVH00010, EMD Millipore). Membranes were blocked with 5% non-fat milk and incubated with the following primary antibodies: β-actin (1:2000; A5316, Sigma-Aldrich, 1:1000; A4700, Sigma-Aldrich), Lamin B1 (1:1000; ab16048, Abcam), p16[INK4a] (1:1000; sc-56330, Santa Cruz, 1:1000; 11104, IBL), and p21[Waf1/Cip1] (1:1000; 2947, Cell Signaling Technology). After washing, membranes were incubated with HRP-conjugated secondary antibodies (1:2000; 7074 and 7076, Cell Signaling Technology) and developed using Amersham ECL Prime or Select reagents (RPN2236 or RPN2235, GE Healthcare). Chemiluminescent signals were detected using the Amersham ImageQuant 800 system (Cytiva). Uncropped and unprocessed blot images are provided in the Source Data file.

## EdU incorporation assay

EdU incorporation analysis was performed using the Alexa Fluor 555 Click-iT EdU Imaging Kit (C10637, Thermo Fisher Scientific), with the following modifications to the manufacturer's protocol. Cells were incubated with 50 μM EdU for 7 days prior to fixation. Following fixation, two sequential 1-h click reactions were performed using freshly prepared reaction solutions. Coverslips were then washed three times with 3% BSA in PBS. Nuclei were counterstained with Hoechst33342 (H342, Dojindo). EdU-positive signals were acquired using an all-in-one fluorescence microscope (BZ-710, Keyence). Quantification was performed using Fiji (Schindelin et al, 2012).

## Apoptosis assay

Apoptotic cells were detected using annexin V staining or TUNEL staining. For annexin V staining, cells were stained with FITC-conjugated annexin V (A13199, Thermo Fisher Scientific) in binding buffer (10 mM HEPES, 140 mM NaCl, and 2.5 mM CaCl₂, pH 7.4) for 15 min. Nuclei were counterstained with DAPI (340-07971, Dojindo). For TUNEL staining, 4% PFA-fixed cells were processed using the One-step TUNEL In Situ Apoptosis Kit (Red, Elab Fluor 555, E-CA-A325, Elabscience) according to the manufacturer's instructions. Nuclei were counterstained with DAPI (340-07971, Dojindo). Images were acquired using an all-in-one fluorescence microscope (BZ-710, Keyence), and quantification was performed using Fiji (ImageJ).

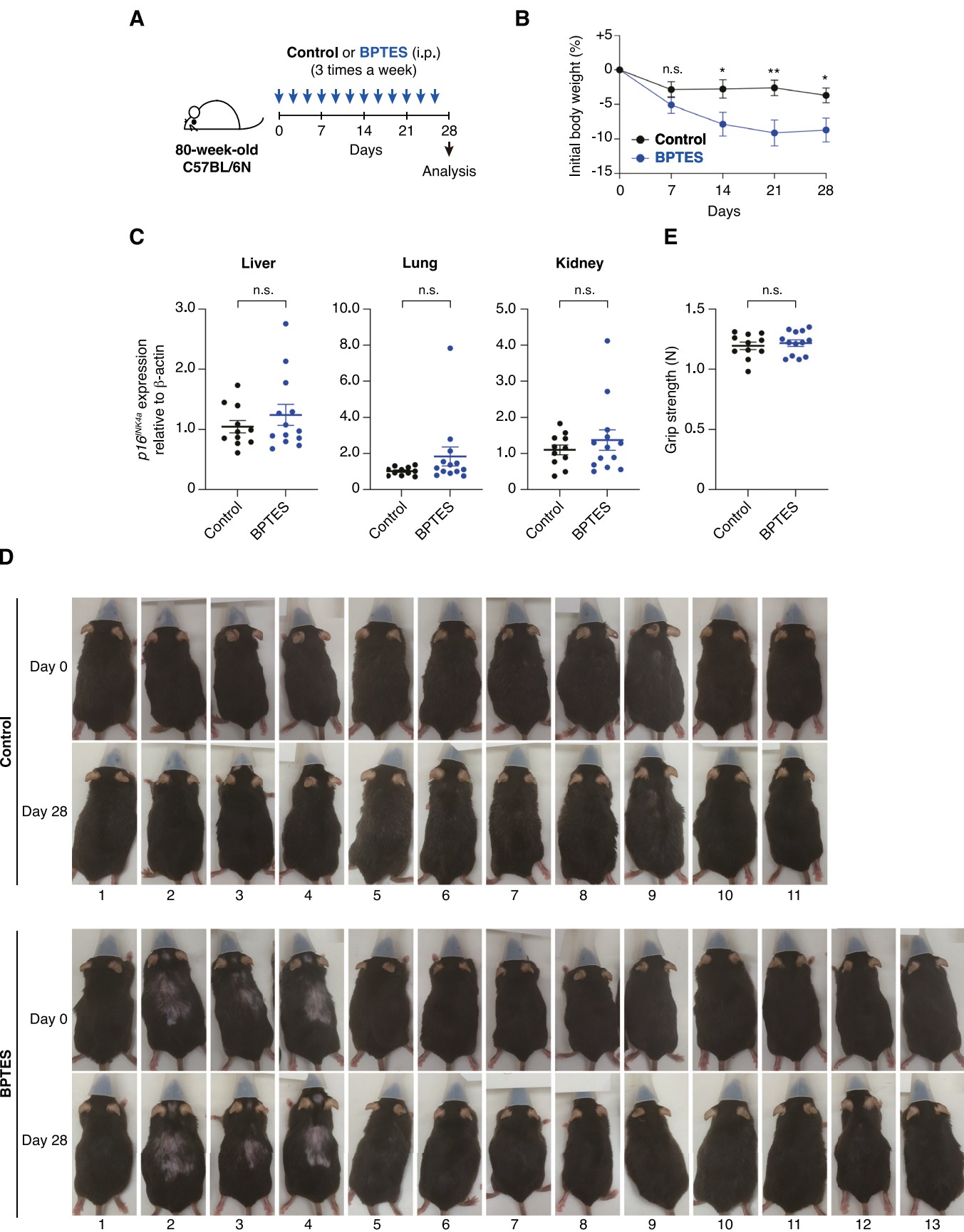

**Figure 3.   Blinded experiments failed to detect senolytic effects of BPTES in aged.**

(A) Timeline of the experimental procedure for BPTES administration in 80-week-old C57BL/6 N mice. (B) Body weight changes over time, shown relative to the starting weight, in mice treated with BPTES or vehicle control (control, $n = 11$; BPTES, $n = 13$). (C) Expression of $p16^{INK4a}$ in the liver, lung, and kidney of 80-week-old mice treated with BPTES or vehicle control, assessed by RT-qPCR (control, $n = 11$; BPTES, $n = 13$). Fold changes were calculated using the ΔΔCT method relative to the vehicle control group. (D) Representative images of all mice before (day 0) and after (day 28) the completion of treatment with BPTES or vehicle control (control, $n = 11$; BPTES, $n = 13$). (E) Grip strength measurements in 80-week-old mice treated with BPTES or vehicle control (control, $n = 11$; BPTES, $n = 13$). Data are presented as mean ± s.e.m. Statistical significance was determined by the Mann–Whitney $U$ test (B, E) and two-tailed unpaired $t$-test (C). Day7, $p = 0.21$; Day14, *$p = 0.022$; Day21, **$p = 0.0088$; Day28, *$p = 0.035$ (B). Liver, $p = 0.36$; Lung, $p = 0.17$; Kidney, $p = 0.42$ (C). $p = 0.64$ (E). n.s., not significant ($p \geq 0.05$). Source data are available online for this figure.

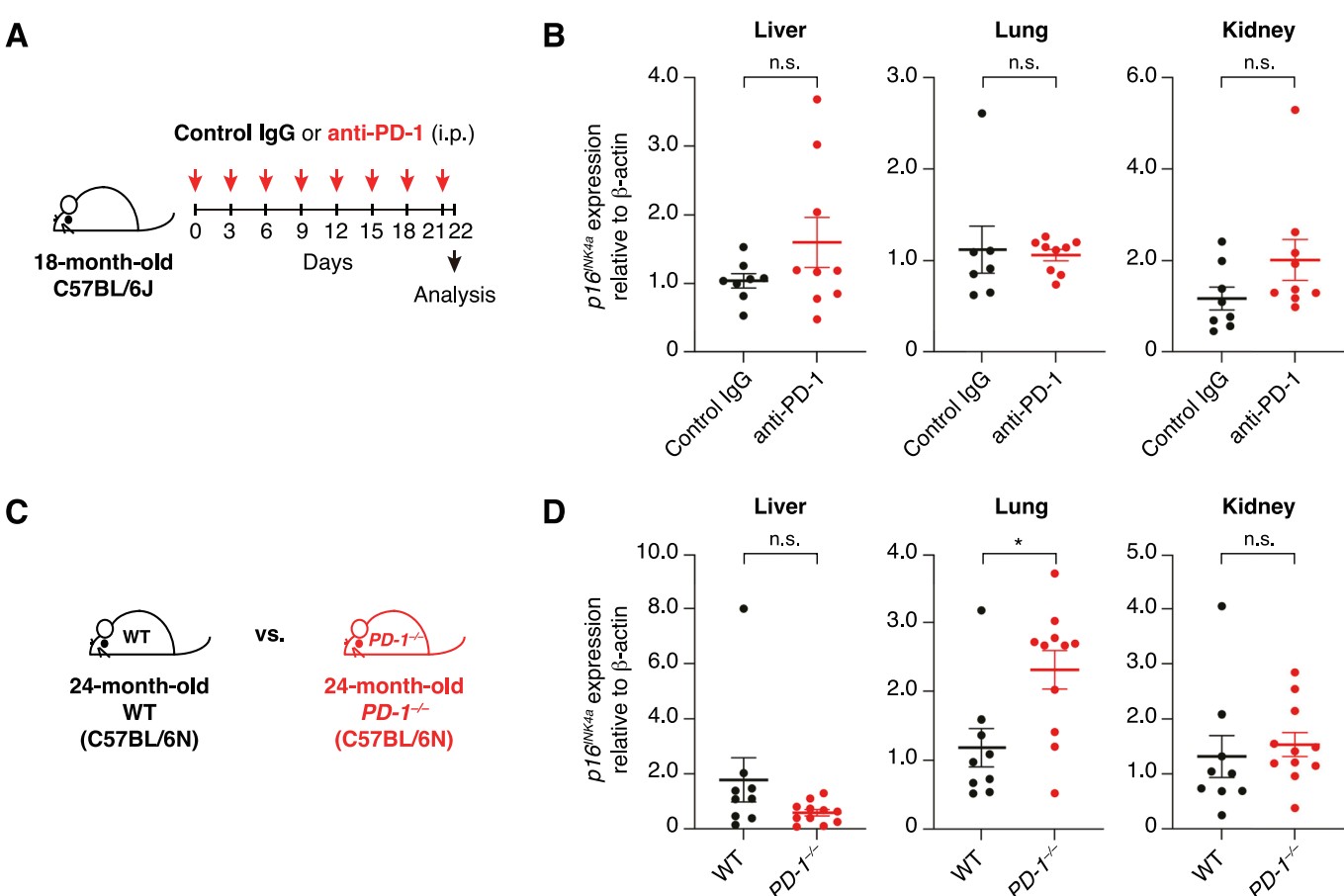

**Figure 4.   PD-1 blockade fails to exhibit senolytic effects in aged mice.**

(A) Timeline of the experimental procedure for anti-PD-1 antibody administration in 18-month-old C57BL/6 J mice. (B) $p16^{INK4a}$ expression in the liver, lung, and kidney of 18-month-old C57BL/6 J mice treated with anti-PD-1 antibody or control IgG (control IgG, $n = 7$–8; anti-PD-1, $n = 9$), assessed by RT-qPCR. Fold changes were calculated using the ΔΔCT method relative to the control IgG group. (C) Schematic of the experimental design involving 24-month-old C57BL/6N-background wild-type (WT) and $PD-1^{-/-}$ mice. (D) $p16^{INK4a}$ expression in the liver, lung, and kidney of 24-month-old C57BL/6N-background WT and $PD-1^{-/-}$ mice (WT, $n = 9$; $PD-1^{-/-}$, $n = 11$), assessed by RT-qPCR. Fold changes were calculated using the ΔΔCT values relative to the WT group. Data are presented as mean ± s.e.m. Statistical significance was determined by two-tailed unpaired $t$-tests (B, D). Liver, $p = 0.18$; Lung, $p = 0.80$; Kidney, $p = 0.14$ (B). Liver, $p = 0.12$; Lung, *$p = 0.011$; Kidney, $p = 0.60$ (D). n.s., not significant ($p \geq 0.05$). Source data are available online for this figure.

## Mice

C57BL/6 J or C57BL/6 N male mice (8-week-old or 80-week-old, and 18-month-old) were purchased from CLEA Japan. Some of the aged mice were provided by the National BioResource Project (NBRP) of the Ministry of Education, Culture, Sports, Science and Technology in Japan (MEXT) through the Foundation for Biomedical Research and Innovation at Kobe, and by the Japan Agency for Medical Research

and Development (AMED) under grant number JP24gm1710001h0003. PD-1 knockout ($PD-1^{-/-}$) (Nishimura et al, 2001), p16/p21 double knockout (DKO) (Kawamoto et al, 2023), and wild-type (WT) mice on a C57BL/6 background were bred and maintained under specific pathogen-free (SPF) conditions at the RIKEN Center for Integrative Medical Sciences (IMS) (AEY2024-009), the Research Institute for Microbial Diseases (RIMD) at the University of Osaka (Biken-AP-R07-05-0), and Kumamoto University (A2024-

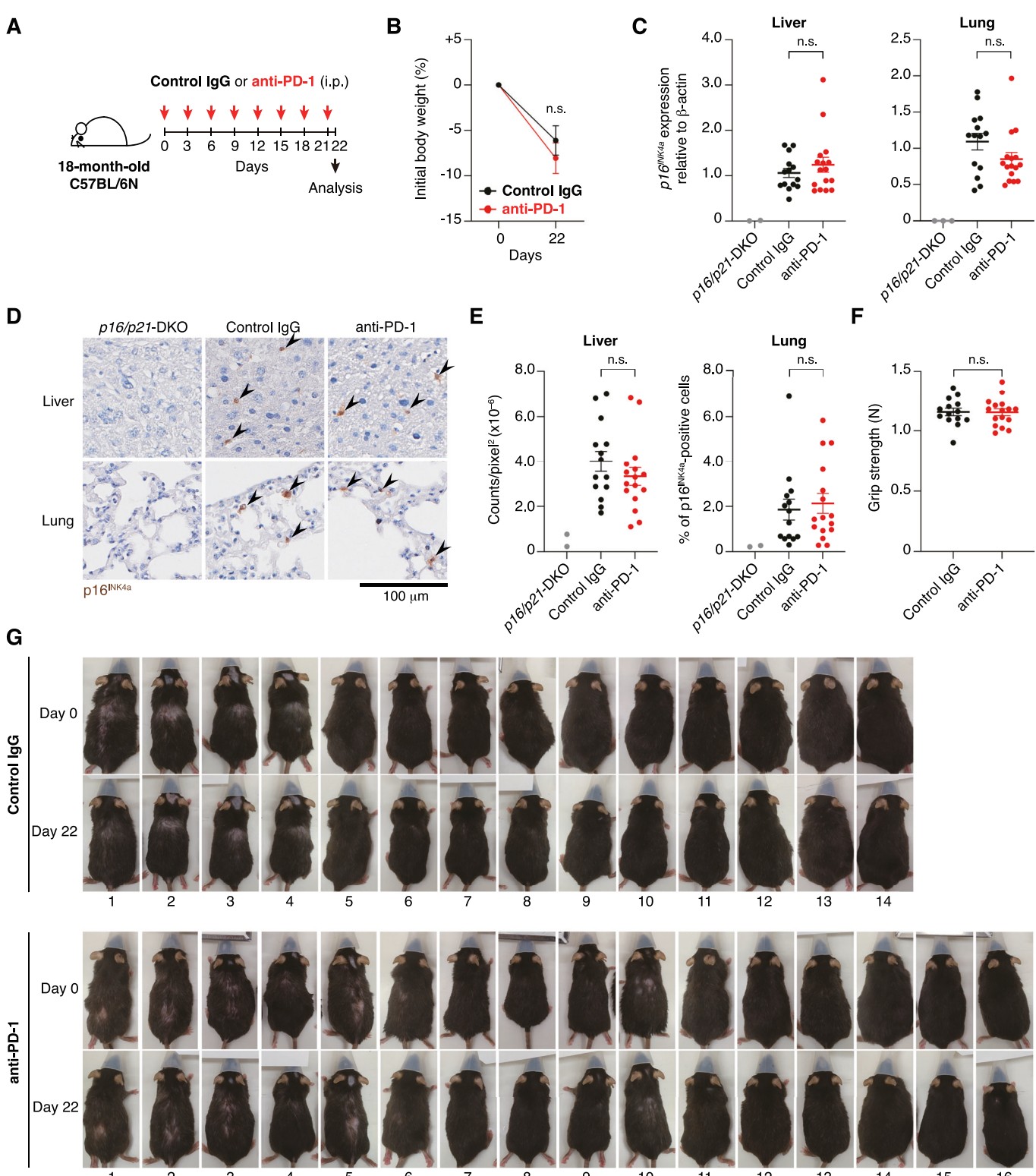

123). All mice were housed in a temperature- (23–25 °C) and humidity-controlled environment on a 12-h light/dark cycle, and were fed standard chow (CE-2; CLEA Japan) sterilized by 20 kGy gamma irradiation. All animal experiments were conducted in accordance with protocols approved by the Animal Research Committees of RIKEN IMS, RIMD at the University of Osaka, or Kumamoto University. Mice were euthanized using carbon dioxide, and all efforts were made to minimize suffering. To minimize potential confounding factors, mice that exhibited tumor development were excluded from all downstream analyses.

**Figure 5.  Blinded experiments failed to detect senolytic effects of PD-1 blockade in aged mice.**

(A) Timeline of the experimental procedure for anti-PD-1 antibody administration in 18-month-old C57BL/6 N mice. (B) Body weight changes over time, shown relative to the starting weight, in mice treated with anti-PD-1 antibody or control IgG (control IgG, $n = 14$; anti-PD-1, $n = 16$). (C–E) Expression of $p16^{INK4a}$ in the liver and lung assessed by RT-qPCR (C), and representative images (D) and histological quantification of $p16^{INK4a}$-positive cells (E) in liver and lung in 18-month-old C57BL/6 N mice treated with anti-PD-1 antibody or control IgG. For RT-qPCR analysis, fold changes were calculated using the $\Delta\Delta CT$ method relative to the control IgG group. $p16/p21$-DKO mice were included as a negative control (control IgG, $n = 14$; anti-PD-1, $n = 16$; $p16/p21$-DKO, $n = 2$–3). (F) Grip strength measurements in 18-month-old C57BL/6 N mice treated with anti-PD-1 antibody or control IgG (control IgG, $n = 14$; anti-PD-1, $n = 16$). (G) Representative images of all mice before (day 0) and after (day 22) the completion of treatment with anti-PD-1 antibody or control IgG (control IgG, $n = 14$; anti-PD-1, $n = 16$). Data are presented as mean ± s.e.m. Statistical significance was determined by the Mann–Whitney $U$ test (B, E, F) and two-tailed unpaired $t$-test (C). The $p16/p21$-DKO samples were employed solely as negative controls and were excluded from statistical analyses (C–E). $p = 0.45$ (B). Liver, $p = 0.37$; Lung, $p = 0.11$ (C). Liver, $p = 0.33$; Lung, $p = 0.70$ (E). $p = 0.89$ (F). n.s., not significant ($p \geq 0.05$). Source data are available online for this figure.

## BPTES administration

BPTES administration was performed as previously described in Johmura et al (2021). Male C57BL/6 J or C57BL/6 N mice (8 or 80 weeks old) were intraperitoneally injected with BPTES (SML0601; Sigma-Aldrich) at a dose of 0.25 mg per 20 g body weight in 200 µl volume, or with a vehicle control (200 µl of 10% DMSO in corn oil), three times per week for one month. In certain experiments, randomization of samples was performed, and the procedures were carried out in a blinded fashion by different experimental facilities and personnel.

## Anti-PD-1 antibody administration

Anti-PD-1 antibody administration was performed as previously described in Wang et al (2022). Eighteen-month-old male C57BL/6 J or C57BL/6 N mice were intraperitoneally injected with 250 µg of either isotype control IgG (clone RTK2758; 400563, BioLegend) or anti-mouse PD-1 antibody (clone 29F.1A12; 135234, BioLegend) every 3 days for 3 weeks. Mice were euthanized one day after the final injection. In certain experiments, randomization of samples was performed, and the procedures were carried out in a blinded fashion by different experimental facilities and personnel.

## Measurement of grip strength

Forelimb grip strength (in newtons, N) was measured using a grip strength meter (Columbus Instruments, Columbus, OH). Each mouse was gently held by the base of the tail and allowed to grasp a horizontal metal bar with its forepaws. The mouse was then pulled backward in a horizontal plane until it released the bar. The peak force exerted before release was recorded as the grip strength. Each mouse was tested three consecutive times, and the highest value among the three trials was used for analysis. All measurements were performed by the same experimenter.

## Immunohistochemistry

Tissue specimens were fixed in Bouin's fixative (33142, Muto Pure Chemicals) for 2 h at room temperature, soaked overnight in 70% ethanol, embedded in paraffin, and sectioned at a thickness of 5 µm using a microtome (Leica Biosystems). The sections were deparaffinized in PathoClean (161-28321, FUJIFILM Wako Pure Chemical Corporation), rehydrated, and subjected to antigen retrieval by microwave heating in 10 mM citrate buffer (pH 6.0)

for 20 min. After blocking with 5% horse serum, the sections were treated with 2% hydrogen peroxide for 10 min to inactivate endogenous peroxidase activity. Anti-mouse $p16^{INK4a}$ (1:1000; ab211542, Abcam) was used as the primary antibody (Doolittle et al, 2023; Kawamoto et al, 2023). Signal detection was performed using the ImmPRESS HRP horse anti-rabbit IgG polymer detection kit (MP-7401, Vector Laboratories) and the ImmPACT DAB substrate kit (SK-4105, Vector Laboratories). Finally, the sections were counterstained with hematoxylin (30002, Muto Pure Chemicals), dehydrated, and mounted with PARAmount (308-400, FALMA).

## Image processing and analysis for $p16^{INK4a}$-positive cell quantification

Tissue sections were digitally scanned using SLIDEVIEW VS200 Universal Whole Slide Imaging Scanner (Evident) equipped with a 20× objective lens. The resulting images were converted to TIFF format and processed using ImageJ Fiji (Schindelin et al, 2012). Immunohistochemical staining was quantified using a custom macro that performed color deconvolution to separate the images into three components: hematoxylin (nuclei), DAB (positive immunostaining), and residual staining. Each separated channel was saved as an individual TIFF file. For lung tissue samples, a MATLAB-based pipeline incorporating Cellpose was used (Stringer et al, 2021; MATLAB R2024a, The MathWorks, Inc.). The workflow involved processing each image with Cellpose, and quantifying total cells and DAB-positive cells. For liver tissue samples, we used a custom ImageJ macro. First, hematoxylin-stained images were used to define tissue boundaries. These images were subjected to Gaussian blurring, followed by binary thresholding to generate tissue masks. DAB-positive cell counting was conducted using the "Find Maxima" function of Fiji.

## Quantitative real-time PCR analysis

Total RNA was extracted from tissues using the RNeasy Mini Kit (74106, QIAGEN) according to the manufacturer's instructions. A total of 1000 ng of RNA was used to synthesize cDNA with the PrimeScript RT Reagent Kit with gDNA Eraser (RR047A, Takara Bio). Quantitative real-time PCR was performed using the Thermal Cycler Dice Real Time System III (Takara Bio) and the TB Green Premix Ex Taq II (RR820A, Takara Bio). The mRNA expression level of each gene was normalized to β-actin (*Actb*) expression. Relative expression levels were calculated using the $\Delta\Delta CT$ method. The specific primers used were as follows:

- *Actb*: 5′-GATGACCCAGATCATGTTTGA-3′ and 5′-GGAGAG-CATAGCCCTCGTAG-3′
- *p16^INK4a^*: 5′-GAACTCTTTCGGTCGTACCC-3′ and 5′-CGAATCTGCACCGTAGTTGA-3′

## Statistical analysis

The graphs or plots are presented as mean ± SD (standard deviation) or mean ± SEM (standard error of the mean) with sample size ≥3 and indicated in the figure legend. The datasets were not tested for normality. All data were visualized and analyzed using Prism (version 10.4.0; GraphPad Software). Statistical significance was assessed using the one-way ANOVA with Dunnett's test (Fig. 1A,C), Mann–Whitney U test (Figs. 2B, 3B,E and 5B,E,G), one-way ANOVA followed by Šídák's multiple comparisons test (Fig. 2C), or a two-tailed unpaired *t*-test (Figs. 3C, 4B,D, and 5C). Sample size estimation was not performed.

## Data availability

This study includes no data deposited in external repositories.

The source data of this paper are collected in the following database record: biostudies:S-SCDT-10_1038-S44319-026-00740-5.

## Peer review information

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

## Acknowledgements

We thank Drs. Tasuku Honjo and Sidonia Fagarasan for providing us with the *PD-1*-deficient mice. We also thank Yuko Wakabayashi for assistance with tissue staining. We are grateful to all members of the laboratories of Takahashi, Kondoh, Oike, Sugimoto, Matsuda, Mochizuki, and Hara for their valuable discussions during the preparation of this manuscript. Some of the aged mice were provided by the National BioResource Project (NBRP) of the Ministry of Education, Culture, Sports, Science and Technology in Japan (MEXT) through the Foundation for Biomedical Research and Innovation at Kobe, and by the Japan Agency for Medical Research and Development (AMED) under grant number JP24gm1710001h0003. This work was supported in part by grants from the Japan Science and Technology Agency (JST) under grant number JPMJFR2308 (to SK), AMED under grant number JP24gm1710004h0003 (to EH) and JP25zf0127008h0002 (to EH), Japan Society for the Promotion of Science (JSPS) under grant number JP23K06481 (to SK) and JP25H00443 (to EH), MEXT under grant number JPMXP1323015484 (to EH), the Takeda Science Foundation (to SK), the LOTTE Foundation (to SK), the Nagase Science and Technology Foundation (to SK), Mitsubishi Foundation (to EH) and from OU Master Plan Implementation Project promoted under the University of Osaka (to EH). A part of the figures was created with BioRender.com (https://www.biorender.com/). Finally, we dedicate this work to the memory of Dr. Yoshimi Takai, who believed in the importance of this study. His steadfast encouragement gave us the courage to undertake this difficult research that many hesitated to pursue. We are profoundly grateful for his mentorship.

## Author contributions

**Shimpei Kawamoto**: Conceptualization; Data curation; Formal analysis; Funding acquisition; Investigation; Visualization; Methodology; Writing—review and editing. **Haruki Horiguchi**: Data curation; Formal analysis; Investigation; Visualization; Methodology; Writing—review and editing. **Daisuke Torigoe**: Data curation; Formal analysis; Investigation; Methodology; Writing—review and editing. **Masahiro Wakita**: Data curation; Formal analysis; Investigation; Visualization; Methodology; Writing—review and editing. **Koyu Ito**: Data curation; Formal analysis; Investigation; Visualization; Methodology; Writing—review and editing. **Sho Sugawara**: Data curation; Formal analysis; Investigation; Visualization; Methodology; Writing—review and editing. **Xiangyu Zhou**: Data curation; Formal analysis; Investigation; Visualization; Methodology; Writing—review and editing. **Takumi Mikawa**: Data curation; Formal analysis; Investigation; Visualization; Methodology; Writing—review and editing. **Jeong Hoon Park**: Formal analysis; Investigation; Methodology; Writing—review and editing. **Birte Kristin Jung**: Investigation; Writing—review and editing. **Yumiko Okumura**: Investigation; Writing—review and editing. **Hideka Miyagawa**: Resources; Writing—review and editing. **Mikako Maruya**: Resources; Writing—review and editing. **Nozomi Hori**: Investigation; Writing—review and editing. **Ken Uemura**: Investigation; Writing—review and editing. **Masataka Sugimoto**: Methodology; Writing—review and editing. **Michiyuki Matsuda**: Data curation; Formal analysis; Investigation; Writing—review and editing. **Naoki Mochizuki**: Conceptualization; Supervision; Writing—review and editing. **Hiroshi Kondoh**: Conceptualization; Formal analysis; Supervision; Methodology; Writing—review and editing. **Akiko Takahashi**: Conceptualization; Data curation; Formal analysis; Supervision; Methodology; Writing—review and editing. **Yuichi Oike**: Conceptualization; Data curation; Formal analysis; Supervision; Methodology; Writing—review and editing. **Eiji Hara**: Conceptualization; Data curation; Formal analysis; Supervision; Funding acquisition; Methodology; Writing—original draft; Project administration; Writing—review and editing.

Source data underlying figure panels in this paper may have individual authorship assigned. Where available, figure panel/source data authorship is listed in the following database record: biostudies:S-SCDT-10_1038-S44319-026-00740-5.

## Disclosure and competing interests statement

MS is a co-author of the Science paper (Science. 2021 Jan 15;371(6526):265–270), which is critically discussed in the present manuscript. In that study, MS's contribution was limited to administering the GLS1 inhibitor to aged mice. In the present study, MS provided technical advice on GLS1 inhibitor administration based on his prior experience. MS was not involved in the design, execution, or interpretation of the phenotypic or molecular analyses in either study. The other authors declare no competing interests.

# Expanded View Figures

**Figure EV1.  High dose of BPTES elicit apoptosis in control HDFs.**

(**A–D**) Non-senescent early-passage HDFs, IMR-90 (**A**, **C**) and TIG-3 (**B**, **D**), were treated with BPTES from Sigma-Aldrich or Cayman at the indicated concentrations for 2 days. Apoptotic cells positive for fluorochrome-labeled annexin V or TUNEL were visualized by fluorescence microscopy, and the proportions of annexin V- or TUNEL-positive cells were quantified. Data are presented as mean ± s.d. (**A–D**, $n = 4$). All of the experiments were repeated at least twice, independently, with similar results. Statistical significance was determined by one-way ANOVA followed by Sidak's test. DMSO vs. BPTES (Sigma) 10 mM: $**p = 0.0018$ (IMR-90), $***p = 0.0002$ (TIG-3); DMSO vs. BPTES (Cayman) 10 μM: $****p < 0.0001$ (IMR-90, TIG-3); BPTES (Sigma) 1 vs. 10 μM: $**p = 0.0016$ (IMR-90), $***p = 0.0002$ (TIG-3); BPTES (Cayman) 1 vs. 10 μM: $****p < 0.0001$ (IMR-90, TIG-3) (**A**, **B**). DMSO vs. BPTES (Sigma) 10 μM: $***p = 0.0001$ (IMR-90), $****p < 0.0001$ (TIG-3); DMSO vs. BPTES (Cayman) 10 μM: $***p = 0.0002$ (IMR-90), $***p = 0.0006$ (TIG-3); BPTES (Sigma) 1 vs. 10 μM: $***p = 0.0003$ (IMR-90), $***p = 0.0001$ (TIG-3); BPTES (Cayman) 1 vs. 10 μM: $***p = 0.0003$ (IMR-90), $***p = 0.0006$ (TIG-3) (**C**, **D**). Scale bars, 10 μm. Source data are available online for this figure.

▶

**A**  **IMR-90 cells**

DMSO
(vehicle)

BPTES (Sigma)
1 µM    10 µM

BPTES (Cayman)
1 µM    10 µM

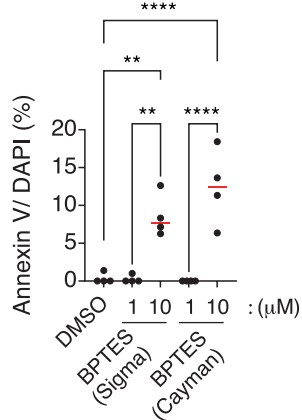

Annexin V  DAPI

**B**  **TIG-3 cells**

DMSO
(vehicle)

BPTES (Sigma)
1 µM    10 µM

BPTES (Cayman)
1 µM    10 µM

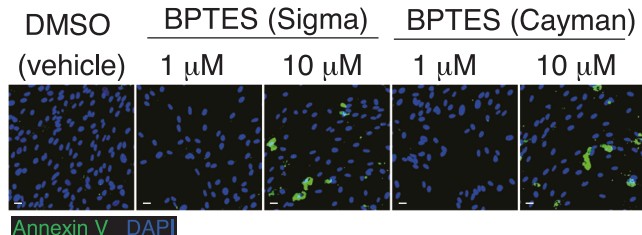
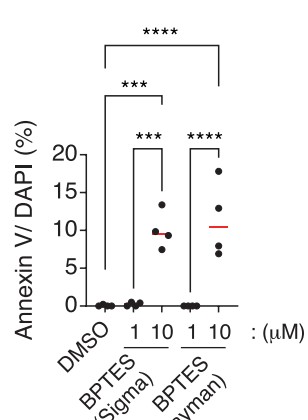

Annexin V  DAPI

**C**  **IMR-90 cells**

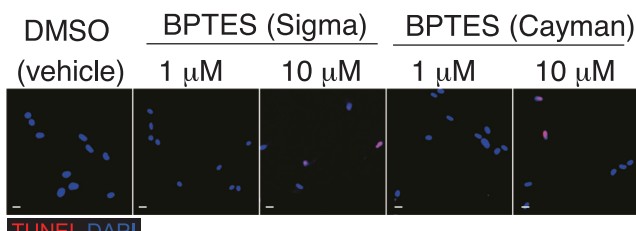
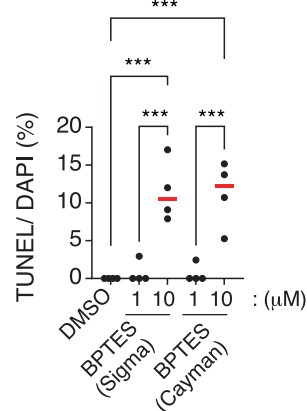

TUNEL DAPI

**D**  **TIG-3 cells**

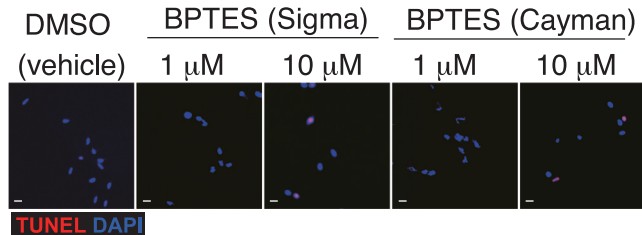
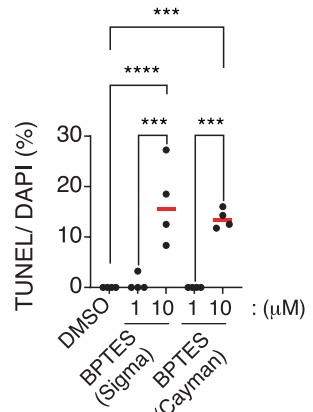

TUNEL DAPI

## A

### IMR-90 cells

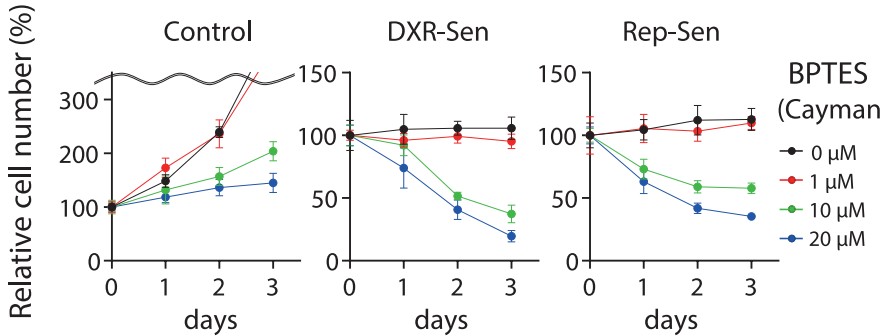

## B

### TIG-3 cells

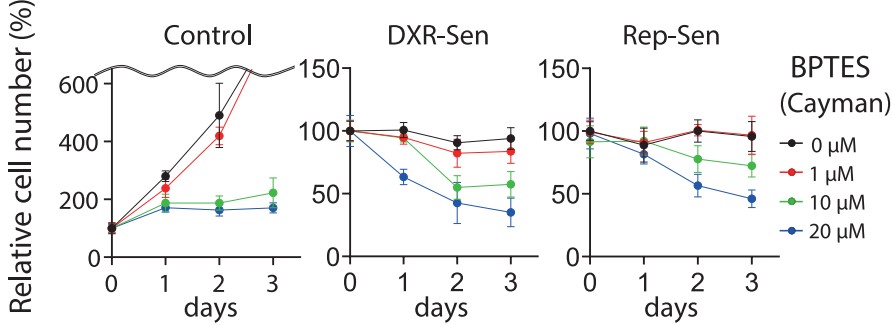

## C

### TIG-1 cells

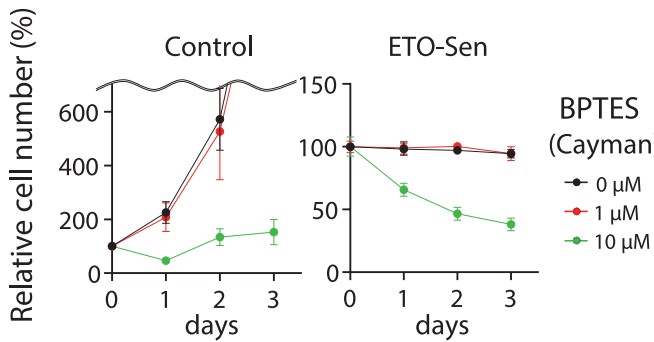

**Figure EV2. Effect of BPTES from different suppliers on proliferation and survival of senescent HDFs.**

(A–C) Control and senescent HDFs were treated with BPTES from Cayman at the indicated concentrations for 3 days. Cell numbers were monitored throughout the experimental period, and relative cell numbers were quantified. Data are presented as mean ± s.d. (A, B: n = 4; C: n = 3). All of the experiments were repeated at least twice, independently, with similar results. Source data are available online for this figure.

