## [Peer Review File · EMBO Reports]

Reevaluating the Senolytic Activity of a GLS1 Inhibitor and an Anti-PD-1 Antibody: Toward Greater Reproducibility and Methodological Rigor

Shimpei Kawamoto, Haruki Horiguchi, Daisuke Torigoe, Masahiro Wakita, Koyu Ito, Sho Sugawara, Xiangyu Zhou, Takumi Mikawa, Jeong Hoon Park, Birte Jung, Yumiko Okumura, Hideka Miyagawa, Mikako Maruya, Nozomi Hori, Ken Uemura, Masataka Sugimoto, Michiyuki Matsuda, Naoki Mochizuki, Hiroshi Kondoh, Akiko Takahashi, Yuichi Oike, and Eiji Hara

Corresponding author(s): Eiji Hara (ehara@biken.osaka-u.ac.jp) , Shimpei Kawamoto (kshimpei@biken.osaka-u.ac.jp)

Review Timeline:

Submission Date:	14th May 25
Editorial Decision:	29th Aug 25
Revision Received:	7th Oct 25
Editorial Decision:	19th Jan 26
Revision Received:	22nd Jan 26
Accepted:	13th Feb 26

Transaction Report:

Dear Eiji,

Thank you for the submission of your research manuscript to our journal. As you know, we have invited a response from Dr. Nakanishi and both articles were reviewed side by side.

We have now received the full set of referee reports for your manuscript that is copied below.

As you will see, all three referees acknowledge the importance of independent validation experiments and the assessment of reproducibility in the senolytics field and beyond and consider your work an important contribution, providing insight into the context-dependent responses to senotherapeutics.

That said, the referees also raise several largely overlapping concerns, the most significant being:

- 1) The limited analysis of senescence that is largely based on the evaluation of p16 expression.
- 2) The absence of a more extensive healthspan analysis that would add support for the grip strength experiments provided.

Referee #3 further comments on the set-up of your study, being not a multi-center analysis in its true sense, which would require parallel experimentation in different labs. I tend to agree with this assessment and suggest to re-phrase the title and/or be more explicit that experimentation and analysis were performed in different laboratories.

Given the positive evaluation by the referees and the value of your data for the field, I would like to invite you to revise your study for publication in EMBO Reports. Further in vivo experiments to analyse more parameters of healthspan or to analyse mice older than 18 months are not required at this stage, but I would encourage the use and analysis of existing samples, if available, to probe for alternative senescence markers. Please also address all other concerns and take the suggestions of the referees on board regarding a maybe even more careful phrasing and an extension of the discussion of potential reasons of discrepancy, potential similarities, and the limitations of your study (such as the absence of a more in-depth health-span analysis). I think it is also a good suggestion from referee #1 to extend the discussion a bit more beyond the refuting aspect.

Please let me know in case you disagree, and we can discuss the exact revision requirements further, also in a video chat, if you like.

It is EMBO Reports policy to allow a single round of revision only and acceptance or rejection of the manuscript will therefore depend on the completeness of your responses included in the next, final version of the manuscript.

We realize that it is difficult to revise to a specific deadline. In the interest of protecting the conceptual advance provided by the work, we recommend a revision within 3 months (November 28th). Please discuss the revision progress ahead of this time with the editor if you require more time to complete the revisions.

While your manuscript would fulfil the criteria of a short report, we can also publish it as a full article to give enough room for a separate discussion section.

=====

IMPORTANT NOTE:

We perform an initial quality control of all revised manuscripts before re-review. Your manuscript will FAIL this control and the handling will be delayed IN CASE the following APPLIES:

- 1) A data availability section providing access to data deposited in public databases is missing. If you have not deposited any data, please add a sentence to the data availability section that explains that.
- 2) Your manuscript contains statistics and error bars based on $n=2$. Please use scatter blots in these cases. No statistics should be calculated if $n=2$.

=====

2) individual production quality figure files as .eps, .tif, .jpg (one file per figure). Please download our Figure Preparation Guidelines (figure preparation pdf) from our Author Guidelines pages <https://www.embopress.org/page/journal/14693178/authorguide> for more info on how to prepare your figures.

4) a complete author checklist, which you can download from our author guidelines (). Please insert information in the checklist that is also reflected in the manuscript. The completed author checklist will also be part of the RPF.

5) Please note that all corresponding authors are required to supply an ORCID ID for their name upon submission of a revised manuscript (). Please find instructions on how to link your ORCID ID to your account in our manuscript tracking system in our Author guidelines
()

6) We replaced Supplementary Information with Expanded View (EV) Figures and Tables that are collapsible/expandable online. A maximum of 5 EV Figures can be typeset. EV Figures should be cited as 'Figure EV1, Figure EV2" etc... in the text and their respective legends should be included in the main text after the legends of regular figures.

7) Please include a dedicated "Data Availability" section at the end of the Methods (suggested wording: "The [structural coordinates | microarray | mass spectrometry] data from this publication have been deposited to the [name of the database] database [URL] and assigned the identifier [accession | permalink | hashtag]."). Should this not apply, this should still be stated as "This study includes no data deposited in external repositories."

Additional information on source data and instruction on how to label the files are available

10) Figure legends and data quantification:
The following points must be specified in each figure legend:

- the name of the statistical test used to generate error bars and P values,
- the EXACT p-values,
- the number (n) of independent experiments (please specify technical or biological replicates) underlying each data point,
- the nature of the bars and error bars (s.d., s.e.m.)

- If the data are obtained from n {less than or equal to} 5, show the individual data points in addition to the SD or SEM.
- If the data are obtained from n {less than or equal to} 2, use scatter blots showing the individual data points.

Discussion of statistical methodology can be reported in the materials and methods section, but figure legends should contain a

basic description of n, P and the test applied.

11) Our journal encourages inclusion of *data citations in the reference list* to directly cite datasets that were re-used and obtained from public databases. Data citations in the article text are distinct from normal bibliographical citations and should directly link to the database records from which the data can be accessed. In the main text, data citations are formatted as follows: "Data ref: Smith et al, 2001" or "Data ref: NCBI Sequence Read Archive PRJNA342805, 2017". In the Reference list, data citations must be labeled with "[DATASET]". A data reference must provide the database name, accession number/identifiers and a resolvable link to the landing page from which the data can be accessed at the end of the reference. Further instructions are available at .

12) All Materials and Methods need to be described in the main text using our 'Structured Methods' format. According to this format, the Methods section includes a Reagents and Tools Table (listing key reagents, experimental models, software and relevant equipment and including their sources and relevant identifiers) followed by a Methods and Protocols section describing the methods, ideally using a step-by-step protocol format. The aim is to facilitate adoption of the methodologies across labs. Please download and fill our Reagents and Tools Table template (.docx), which you can find in our author guidelines: <https://www.embopress.org/page/journal/14693178/authorguide#structuredmethods>.

13) As part of the EMBO publication's Transparent Editorial Process, EMBO Reports publishes online a Review Process File to accompany accepted manuscripts. This File will be published in conjunction with your paper and will include the referee reports, your point-by-point response and all pertinent correspondence relating to the manuscript.

Kind regards,

Martina

=====

Referee #1:

This is a timely manuscript that I believe should be published after revisions. In fact, more studies like this are needed to address the issue of inter-lab reproducibility and to understand what causes variations in experimental outcomes. This is especially relevant in the senolytic field, where treatment regimens and timing vary widely across both in vitro and in vivo studies. As senolytics move toward clinical application, reproducibility and independent validation become even more critical.

In this multicenter study, the authors tested two compounds, a GLS1 inhibitor (BPTES) and an anti-PD-1 antibody, that were previously reported to reduce senescent cells and improve health in aged mice. However, the authors did not observe the same beneficial effects. They conclude that these findings highlight the importance of rigorous experimental design and independent replication before translating senolytic therapies to the clinic.

The authors attempted to reproduce results from a 2021 Science paper that showed BPTES had senolytic activity and health benefits in aged mice. In their hands, BPTES killed a significant percentage of senescent cells but also showed growth-inhibitory effects in non-senescent cells. This is an interesting observation but does not disprove BPTES's senolytic activity. Most senolytic studies assess short-term cell death and do not focus on long-term effects on proliferating cells. More experiments are needed before ruling out BPTES as a senolytic. For example, Annexin V staining was performed only in proliferating cells, why wasn't this assay also done in senescent cells? A direct comparison of Annexin V staining (or other measures of cell death) between proliferating and senescent cells would be far more informative.

The authors also suggest that BPTES is not senolytic in vivo. Aged mice treated with BPTES lost weight, but no changes were seen in grip strength or p16 mRNA levels. Figure 3D includes images of the mice, presumably to show no visible differences in appearance or coat color. However, the data presented do not allow for strong conclusions. The results differ from the 2021 study, but the authors do not offer any clear explanation for these discrepancies. This should be discussed. Without the context of the previous study, I would say that: Measuring only p16 (in bulk by qPCR) is not enough to evaluate senescence, other markers such as p21 and SASP factors (and others- DDR markers, absence of laminb1) should be assessed. Based on their current data, the authors have not convincingly shown that BPTES lacks senolytic activity. Healthspan evaluation is limited. Only grip strength was measured, and images of mice are not informative without quantification. A more thorough frailty assessment (e.g., following the Whitehead protocol) would have strengthened the conclusions.

In my experience, responses to senotherapeutics in aged mice depend heavily on cohort variability, treatment timing, sex of the animals, and importantly baseline senescent cell burden which varies from cohort to cohort. These factors can significantly affect outcomes, and rescuing age-related phenotypes is not straightforward. The authors state that "the reasons underlying the discrepancy between our results and those reported by Johmura et al. (2021) remain unclear." However, it would be helpful for the reader if they offered possible explanations and perhaps be more upfront about the experimental differences between the studies (was everything exactly the same? What was different?). Additionally, they should acknowledge more clearly the limitations of their own study, particularly regarding the limited senescence and healthspan characterization. That said, I appreciate the authors' emphasis on the importance of independent validation and cautious interpretation of senolytic effects in mice.

Similar points apply to the anti-PD-1 antibody treatment. In this case, senescence was assessed by both qPCR and immunohistochemistry (IHC). I would encourage the authors to include representative images of the p16 staining. While it's reassuring that p16 staining is absent in the p16/p21 double knockout mice, p16 antibodies have historically shown inconsistent results in mouse tissues (most commercial antibodies are not specific). Seeing the actual staining would help evaluate the quality and specificity.

However, the same limitations discussed earlier remain, namely, the lack of comprehensive characterization of both senescence markers and healthspan parameters. The authors were cautious in their conclusions, but I believe the findings should still be described as preliminary.

Ideally, the manuscript would benefit from a more thorough analysis of senescence and healthspan. If conducting additional experiments is not feasible, then the authors should more explicitly acknowledge the limitations of their approach and provide a more detailed direct comparison between their results and those of previous studies. It would also be helpful to highlight both similarities and differences across studies, and to speculate more on possible reasons for the discrepancies. As noted earlier, I believe this type of independent validation is important and should be encouraged in the field. However, it's equally important that the paper is not perceived as an attempt to discredit earlier work by one specific group. In my view, the differences in outcomes can likely be explained by variables such as mouse cohort differences, baseline senescence burden, timing of treatment, and other subtle experimental factors.

Referee #2:

Eijihara et al. report that concerns remain regarding the reproducibility and generalizability of previously reported senolytic effects. In this multicenter study, they rigorously examined the senolytic efficacy of a GLS1 inhibitor and an anti-PD-1 antibody-agents that were previously reported to reduce senescent cell burden and improve health outcomes in aged mice. Contrary to those earlier reports, they observed that neither GLS1 inhibition nor PD-1 blockade significantly reduced senescent cell markers or improved aging-related parameters. These findings underscore the importance of rigorous experimental design, standardized protocols, and independent validation in advancing senolytic strategies toward clinical translation.

I agree with the authors that careful experimental design and independent replication are essential for senolytic research-as they are for all areas of science. However, with decades of collective experience in the life sciences, we recognize that biological processes are inherently complex and that reproducibility across laboratories can be challenging. Differences in cell types, experimental conditions, and readouts can lead to divergent outcomes. In many cases, resolving such discrepancies requires additional studies by multiple groups over time, which is a natural part of scientific progress.

Regarding this specific debate, the original study by Johmura et al. (2021) demonstrated senolytic activity of a GLS1 inhibitor in hHAC2 human skin fibroblasts, whereas Eijihara et al. used other fibroblast types in their replication attempts. Considering the heterogeneity of senescent cells, it is plausible that GLS1 inhibition exhibits cell-type-dependent senolytic effects. Both positive and negative results contribute to our understanding of the complexity and heterogeneity of senescence biology.

Therefore, it may be more constructive to frame the current findings not simply as an inability to reproduce prior results, but rather as evidence that senolytic responses can be heterogeneous and context-dependent. Moreover, the *in vivo* studies presented in Eijihara et al. primarily measured molecular markers (e.g., p16INK4a expression) without fully assessing functional tissue or systemic phenotypes, suggesting that interpretations regarding the absence of physiological effects should be made with caution.

Overall, this work provides valuable insights that highlight the challenges of identifying pan-senolytics and the need for nuanced evaluation of senolytic efficacy across diverse biological contexts. Both positive and negative findings will ultimately help refine the field and advance our understanding of senescence-targeted interventions.

Referee #3:

Reproducibility is at the essence of science but has so far seldom been rigorously tested (i.e. in a well-designed multicentre study) in senolytics research. The Kawamoto et al manuscript is a very laudable approach to this issue. Analysing the impact of two candidate senolytics, BPTES and anti-PD1 in 3 independent laboratories, the authors essentially confirm each others and earlier *in-vitro* data. However, in contrast to the previous publication, they did not find decreased senescence and age indicators, especially p16 expression in multiple organs and alterations in external appearance after drug treatment in 18 month old mice. While these results are interesting, there are some important shortcomings:

1. Experimental design: The *in-vivo* experiments are not performed as a classical multicentre study but use different labs solely for blinding. Accordingly, they do not really "Assess the Reproducibility of Senolytic Experiments". Intervention experiments should be done in parallel in different labs. At least, conclusions need to be toned down and a different title should be chosen.
2. Ageing *in vivo* biomarkers: Conclusions in the ms are based strongly on whole tissue p16RT-PCR results. As outlined in the comment by Johmura et al, this might not be a sensitive ageing marker due to low expression levels and age-independent expression in mice. External appearance has been successfully used as part of a composite frailty measure but as a single marker it is definitely insensitive. Grip strength frequently also has significant reproducibility problems. In summary, I do not think that the parameters employed here give a sensitive assessment of *in-vivo* ageing.
3. The choice of 18 month old mice might reduce the possible effect size. At 18 months, C57Bl6 are not 'really' old (their typical median lifespan is 25-27 months but ageing rates depend significantly on housing conditions). A possibility could be that the Johmura mice did age faster than the Kawamoto ones, resulting in more discernible effects.
4. Further issues related to the differences between the Johmura and Kawamoto results have in my opinion convincingly dealt with in the response by Johmura et al.

Point-by-point responses to the reviewers' comments

We sincerely thank all the reviewers for their constructive and thoughtful comments, which have been invaluable in improving our manuscript. We believe that addressing the reviewers' concerns has strengthened the study. Below, we provide our point-by-point responses to the reviewers' comments, with our replies presented in plain text. We hope that our responses clarify the issues raised and demonstrate that the manuscript merits further consideration.

Reviewer #1:

This is a timely manuscript that I believe should be published after revisions. In fact, more studies like this are needed to address the issue of inter-lab reproducibility and to understand what causes variations in experimental outcomes. This is especially relevant in the senolytic field, where treatment regimens and timing vary widely across both in vitro and in vivo studies. As senolytics move toward clinical application, reproducibility and independent validation become even more critical.

Response:

We thank the reviewer for recognizing the importance of our work. As the reviewer correctly pointed out, the reproducibility and independent validation of senolytic studies are becoming increasingly critical as senolytics move toward clinical applications. In line with this, our cross-laboratory study rigorously tested the reported senolytic efficacies of a GLS1 inhibitor and an anti-PD-1 antibody, which had previously been described in the literature and media as highly effective in eliminating p16^{INK4a}-positive senescent cells and rejuvenating aged mice. Contrary to these earlier reports, however, we did not observe significant reductions in p16^{INK4a}-positive cell burden or improvements in aging-related health parameters. These results underscore the importance of standardized protocols and independent validation, which our study aimed to address.

In this multicenter study, the authors tested two compounds, a GLS1 inhibitor (BPTES) and an anti-PD-1 antibody, that were previously reported to reduce senescent cells and improve health in aged mice. However, the authors did not

observe the same beneficial effects. They conclude that these findings highlight the importance of rigorous experimental design and independent replication before translating senolytic therapies to the clinic.

The authors attempted to reproduce results from a 2021 Science paper that showed BPTES had senolytic activity and health benefits in aged mice. In their hands, BPTES killed a significant percentage of senescent cells but also showed growth-inhibitory effects in non-senescent cells. This is an interesting observation but does not disprove BPTES's senolytic activity. Most senolytic studies assess short-term cell death and do not focus on long-term effects on proliferating cells. More experiments are needed before ruling out BPTES as a senolytic. For example, Annexin V staining was performed only in proliferating cells, why wasn't this assay also done in senescent cells? A direct comparison of Annexin V staining (or other measures of cell death) between proliferating and senescent cells would be far more informative.

Response:

We deeply apologize for the misunderstanding that arose from our insufficient explanation. We do not intend to claim that BPTES lacks senolytic activity altogether. However, through careful experiments conducted in three independent laboratories and across multiple cell types, including IMR90 cells used by Johmura et al., we were unable to reproduce the nearly complete elimination of senescent cells reported in Johmura et al. 2021 (see Fig. 1C, F, G, H and Fig. S5 in Johmura et al. 2021). At the 10 μ M concentration they employed, approximately 50% of the senescent cells remained viable in our experiments. Moreover, at this concentration, we observed a marked inhibition of control cell proliferation (Fig. 1B, D, F, and EV Fig. 2 in our paper) and, in some cases, even signs of apoptosis (our EV Fig. 1), which we believe are noteworthy. We fully recognize that such discrepancies may sometimes be attributable to subtle, unrecognized differences in experimental conditions, and we have added a statement in the revised manuscript to address this possibility (p.9, lines 187–202). Nevertheless, Johmura et al. (2021) concluded that BPTES-induced senolysis resulted from the intracellular acidification of senescent cells. This interpretation was based on the observation that the effect was abolished when the culture medium was alkalized with NH_4OH (Fig. 1G, H and Fig. S5F in Johmura et al., 2021). In contrast, we were unable to obtain such data at all (Fig. 1G in our paper), which we believe is an important observation. This raises the possibility that the mechanism of BPTES-induced senolysis proposed by Johmura et al. may be condition-dependent rather than universally applicable, a point that warrants caution.

Regarding the reviewer's suggestion that apoptosis should be assessed not only in control (non-senescent) cells but also in senescent cells after BPTES treatment, we agree that a direct comparison might be informative. However, because the very definition of a senolytic drug is one that selectively eliminates senescent cells while sparing non-senescent cells, it is expected that apoptosis would occur in senescent cells, whereas the critical issue is whether apoptosis is also induced in non-senescent cells. Indeed, the occurrence of cell death in senescent cells is already evident from the reduction in cell numbers (Fig. 1B, D, F, and EV Fig. 2 in our paper). Thus, we did not consider it essential to perform a dedicated apoptosis assay in senescent cells. By contrast, in control cells, while the BPTES treatment slowed the rate of proliferation, it did not reduce cell numbers (Fig. 1B, D, F, and EV Fig. 2 in our paper). Therefore, it was unclear whether this effect reflected a general slowing of the cell cycle progression or induction of cell death in a subset of cells, which justified performing apoptosis assays in control cells. Since Annexin V indicates an early event of apoptosis, we agree with the reviewer's comment that it would be important to examine additional apoptotic markers. Therefore, we performed a TUNEL assay, which is known as a marker of late-stage apoptosis, and included the results in Expanded View Fig. 1. These additional data further support the notion that BPTES induces apoptosis in a subset of control cells.

We sincerely apologize again for not making these points clearer in the original submission. We hope that our explanation has addressed the reviewer's concerns and that our study will be seen as a constructive contribution to improving reproducibility in the senolytic research field.

The authors also suggest that BPTES is not senolytic in vivo. Aged mice treated with BPTES lost weight, but no changes were seen in grip strength or p16 mRNA levels. Figure 3D includes images of the mice, presumably to show no visible differences in appearance or coat color. However, the data presented do not allow for strong conclusions. The results differ from the 2021 study, but the authors do not offer any clear explanation for these discrepancies. This should be discussed. Without the context of the previous study, I would say that: Measuring only p16 (in bulk by qPCR) is not enough to evaluate senescence, other markers such as p21 and SASP factors (and others- DDR markers, absence of laminb1) should be assessed. Based on their current data, the authors have not convincingly shown that BPTES lacks senolytic activity. Healthspan evaluation is limited. Only grip

strength was measured, and images of mice are not informative without quantification. A more thorough frailty assessment (e.g., following the Whitehead protocol) would have strengthened the conclusions.

In my experience, responses to senotherapeutics in aged mice depend heavily on cohort variability, treatment timing, sex of the animals, and importantly baseline senescent cell burden which varies from cohort to cohort. These factors can significantly affect outcomes, and rescuing age-related phenotypes is not straightforward. The authors state that "the reasons underlying the discrepancy between our results and those reported by Johmura et al. (2021) remain unclear." However, it would be helpful for the reader if they offered possible explanations and perhaps be more upfront about the experimental differences between the studies (was everything exactly the same? What was different?). Additionally, they should acknowledge more clearly the limitations of their own study, particularly regarding the limited senescence and healthspan characterization. That said, I appreciate the authors' emphasis on the importance of independent validation and cautious interpretation of senolytic effects in mice.

Response:

We fully agree that p16 expression alone should not be regarded as a definitive marker of aging in mice. However, the primary aim of our study was not to establish an accurate measure of aging in mice, but rather to assess the reproducibility of the key findings reported by Johmura et al. (2021) and Wang et al. (2022). These studies suggested that the administration of BPTES (a GLS1 inhibitor) and an anti-PD1 antibody could attenuate aging in elderly mice, findings that have attracted considerable public attention and hold profound societal consequences. Indeed, there are even indications that these agents may soon be adopted as off-label, out-of-pocket "anti-aging" therapies in private practice.

Given this context, we deliberately focused on reproducing only the key datasets presented in those reports. In both studies, p16 expression levels were employed as the principal marker of cellular senescence (see Fig. S13A–C in Johmura et al. 2021; Fig. 4a, Extended Data Fig. 7c,d in Wang et al. 2022), while grip strength was adopted as an indicator of organismal aging (see Fig. 4F in Johmura et al. 2021; Fig. 4a, Extended Data Fig. 7f in Wang et al., 2022). [...] Because photographic evidence of rejuvenation phenotypes can strongly influence public perception, we included photographs of all the mice before and after treatment with either BPTES or the anti-PD1 antibody.

We acknowledge, however, that our initial manuscript did not sufficiently explain why we restricted our analysis to p16 expression, grip strength, and the physical appearance of mice. To address this shortcoming, we have now provided a more detailed explanation in the revised manuscript and additionally discuss possible reasons for the discrepancies with previous studies, including inter-individual variation among aged mice and subtle differences in experimental conditions (pp.9-11, lines 187–246). In the experiments in which aged mice were treated with BPTES, the procedures were performed under the guidance of Dr. Sugimoto, who was directly responsible for these experiments in Johmura et al. (2021) and is a co-author of the present study. We therefore believe that the methodological discrepancies in this specific experiment were minimized to the greatest extent possible. In addition, although the reviewer suggested including other markers such as p21, it is well known that in mice the expression of p21 does not consistently increase with age and that p21 is also expressed in non-senescent cell populations. Therefore, we did not consider it to be a reliable marker for this context.

Similar points apply to the anti-PD-1 antibody treatment. In this case, senescence was assessed by both qPCR and immunohistochemistry (IHC). I would encourage the authors to include representative images of the p16 staining. While it's reassuring that p16 staining is absent in the p16/p21 double knockout mice, p16 antibodies have historically shown inconsistent results in mouse tissues (most commercial antibodies are not specific). Seeing the actual staining would help evaluate the quality and specificity.

Response:

We thank the reviewer for pointing this out and fully agree with their concern. After extensive testing over many years, we identified a reliable method for detecting p16 in mouse tissues; namely, the use of a rabbit monoclonal antibody against mouse p16 (EPR20418) in combination with Bouin's fixative (Kawamoto et al., Nat. Cell Biol. 2023). Although this method may not be universally applicable, it has been shown to be effective in tissues with robust p16 expression. The same antibody has also been successfully used by other groups, including Dr. Sundeep Khosla's laboratory at the Mayo Clinic (Doolittle et al., Nat. Commun. 2023). In line with the reviewer's suggestion, we have now included representative images of the p16 staining in the revised manuscript (Fig. 5D in our paper).

However, the same limitations discussed earlier remain, namely, the lack of comprehensive characterization of both senescence markers and healthspan parameters. The authors were cautious in their conclusions, but I believe the findings should still be described as preliminary. Ideally, the manuscript would benefit from a more thorough analysis of senescence and healthspan. If conducting additional experiments is not feasible, then the authors should more explicitly acknowledge the limitations of their approach and provide a more detailed direct comparison between their results and those of previous studies. It would also be helpful to highlight both similarities and differences across studies, and to speculate more on possible reasons for the discrepancies. As noted earlier, I believe this type of independent validation is important and should be encouraged in the field. However, it's equally important that the paper is not perceived as an attempt to discredit earlier work by one specific group. In my view, the differences in outcomes can likely be explained by variables such as mouse cohort differences, baseline senescence burden, timing of treatment, and other subtle experimental factors.

Response:

We thank the reviewer for this insightful comment. We agree that we focused only on p16 expression, grip strength, and the physical appearance of mice, and did not include a comprehensive characterization of additional aging parameters. We have now clearly stated this limitation in the revised text (pp.10-11, lines 222–246).

In addition, we have expanded our discussion to provide a more direct comparison between our results and those of previous studies, highlighting both similarities and differences (pp.9-11, lines 187–246). We also discuss possible reasons for the discrepancies, including inter-individual heterogeneity in aged mice and subtle variations in experimental conditions. We emphasize the importance of conducting experiments in a blinded manner with a sufficient sample size. Importantly, we wish to stress that our intention is not to discredit earlier work by any specific group, but rather to provide an independent validation, which we believe is valuable for the field as senolytic therapies move toward clinical applications (p.11, lines 243-246).

Reviewer #2:

Eiji hara et al. report that concerns remain regarding the reproducibility and generalizability of previously reported senolytic effects. In this multicenter study, they rigorously examined the senolytic efficacy of a GLS1 inhibitor and an anti-PD-1 antibody-agents that were previously reported to reduce senescent cell burden and improve health outcomes in aged mice. Contrary to those earlier reports, they observed that neither GLS1 inhibition nor PD-1 blockade significantly reduced senescent cell markers or improved aging-related parameters. These findings underscore the importance of rigorous experimental design, standardized protocols, and independent validation in advancing senolytic strategies toward clinical translation.

Response:

We sincerely thank the reviewer for providing a clear summary of our work.

I agree with the authors that careful experimental design and independent replication are essential for senolytic research-as they are for all areas of science. However, with decades of collective experience in the life sciences, we recognize that biological processes are inherently complex and that reproducibility across laboratories can be challenging. Differences in cell types, experimental conditions, and readouts can lead to divergent outcomes. In many cases, resolving such discrepancies requires additional studies by multiple groups over time, which is a natural part of scientific progress.

Response:

We are grateful to the reviewer for these insightful comments. We fully agree that reproducibility across laboratories is inherently challenging due to biological complexity and differences in experimental conditions. To address this concern, we conducted cross-laboratory studies involving multiple groups, which helped independently reproduce the findings previously reported by Johmura et al. (2021) and Wang et al. (2022) on senolytics. Moreover, we included the same cell line (IMR-90) used by Johmura et al. (Fig. S5B,C in Johmura et al. 2021) in our analyses (Fig. 1A,B; EV Fig. 1A in our paper), thereby facilitating direct comparison with published results. We believe these efforts contribute to strengthening the reproducibility and to the cumulative progress of senolytic research.

Regarding this specific debate, the original study by Johmura et al. (2021) demonstrated senolytic activity of a GLS1 inhibitor in hHAC2 human skin fibroblasts, whereas Eijihara et al. used other fibroblast types in their replication attempts. Considering the heterogeneity of senescent cells, it is plausible that GLS1 inhibition exhibits cell-type-dependent senolytic effects. Both positive and negative results contribute to our understanding of the complexity and heterogeneity of senescence biology. Therefore, it may be more constructive to frame the current findings not simply as an inability to reproduce prior results, but rather as evidence that senolytic responses can be heterogeneous and context-dependent.

Response:

We thank the reviewer for pointing this out. Although Johmura et al. mainly used hHAC2 cells, they also examined several fibroblast types, including IMR-90 cells, and reported similar results (Fig. S5B,C in Johmura et al., 2021). However, we were unable to reproduce these findings, even in IMR-90 cells (Fig. 1A,B; EV Fig. 1A in our paper). While we acknowledge that cell-type heterogeneity may contribute to variability in senolytic responses, we believe that the discrepancies between Johmura et al. and our study cannot be fully explained by differences in cell type alone.

Moreover, the in vivo studies presented in Eiji hara et al. primarily measured molecular markers (e.g., p16INK4a expression) without fully assessing functional tissue or systemic phenotypes, suggesting that interpretations regarding the absence of physiological effects should be made with caution.

Response:

We fully agree that p16 expression alone should not be regarded as a definitive marker of aging in mice. However, the primary aim of our study was not to establish an accurate measure of aging in mice, but rather to assess the reproducibility of the key findings reported by Johmura et al. (2021) and Wang et al. (2022). These studies suggested that the administration of BPTES (a GLS1 inhibitor) and an anti-PD1 antibody could attenuate aging in elderly mice—findings that have attracted considerable public attention and hold profound societal consequences. Indeed, there are concerns that these agents may soon be adopted as off-label “anti-aging” therapies in private practice.

Given this context, we deliberately focused on reproducing the key datasets

presented in those reports. In both studies, p16 expression levels were employed as the key marker of cellular senescence (see Fig. S13A–C in Johmura et al. 2021; Fig. 4a, Extended Data Fig. 7c,d in Wang et al. 2022), while grip strength was adopted as an indicator of organismal aging (see Fig. 4F in Johmura et al., 2021; Fig. 4a, Extended Data Fig. 7f in Wang et al. 2022). [...] Because photographic evidence of rejuvenation phenotypes can strongly influence public perception, we included photographs of all the mice before and after treatment with either BPTES or the anti-PD1 antibody.

We acknowledge, however, that our initial manuscript did not sufficiently explain why we restricted our analysis to p16 expression, grip strength, and the physical appearance of mice. To address this shortcoming, we have now provided a more detailed explanation in the revised manuscript and additionally discuss possible reasons for the discrepancies with previous studies, including inter-individual variation among aged mice and subtle differences in experimental conditions (pp.9-11, lines 187–246). Notably, in the experiments in which aged mice were treated with BPTES, the procedures were performed under the guidance of Dr. Sugimoto, who was directly responsible for these experiments in Johmura et al. (2021) and is a co-author of the present study. We therefore believe that the methodological discrepancies in this specific experiment were minimized as much as possible.

Overall, this work provides valuable insights that highlight the challenges of identifying pan-senolytics and the need for nuanced evaluation of senolytic efficacy across diverse biological contexts. Both positive and negative findings will ultimately help refine the field and advance our understanding of senescence-targeted interventions.

Response:

We sincerely thank the reviewer for these insightful comments. We fully agree that both positive and negative findings are important for refining the field, and that careful, context-dependent evaluation is essential for assessing senolytic efficacy, particularly as these therapies move toward clinical applications.

Reviewer #3:

Reproducibility is at the essence of science but has so far seldom been rigorously tested (i.e. in a well-designed multicentre study) in senolytics research. The Kawamoto et al manuscript is a very laudable approach to this issue. Analysing the impact of two candidate senolytics, BPTES and anti-PD1 in 3 independent laboratories, the authors essentially confirm each others and earlier in-vitro data. However, in contrast to the previous publication, they did not find decreased senescence and age indicators, especially p16 expression in multiple organs and alterations in external appearance after drug treatment in 18 month old mice.

Response:

We sincerely thank the reviewer for providing a clear summary of our work.

While these results are interesting, there are some important shortcomings:

1. Experimental design: The in-vivo experiments are not performed as a classical multicentre study but use different labs solely for blinding. Accordingly, they do not really "Assess the Reproducibility of Senolytic Experiments". Intervention experiments should be done in parallel in different labs. A least, conclusions need to toned wown and a different title should be chosen.

Response:

We acknowledge that our study does not constitute a classical multicenter study, as the *in vivo* experiments were distributed among laboratories for blinding purposes rather than conducted in parallel. Accordingly, we will revise the abstract and main text to use the term “cross-laboratory study” instead of “multicenter study.” In addition, to address the reviewer’s concern about the scope and interpretation of our work, we have carefully revised the Discussion to moderate the strength of our conclusions (pp.9-11, lines 187–246).

Regarding the title, while our original version did not include the term “multicenter study,” we agree that a more precise description would be beneficial. We therefore propose the revised title: “*Reevaluating the Senolytic Activity of a GLS1 Inhibitor and an anti-PD1 Antibody: Toward Greater Reproducibility and Methodological Rigor.*” We believe this clarifies the focus of our study and reflects the reviewer’s suggestion.

2. Ageing in vivo biomarkers: Conclusions in the ms are based strongly on whole tissue p16RT-PCR results. As outlined in the comment by Johmura et al, this might not be a sensitive ageing marker due to low expression levels and age-independent expression in mice. External appearance has been successfully used as part of a composite frailty measure but as a single marker it is definitely insensitive. Grip strength frequently also has significant reproducibility problems. In summary, I do not think that the parameters employed here give a sensitive assessment of in-vivo ageing.

Response:

We fully agree that p16 expression alone should not be regarded as a definitive marker of aging in mice. However, the primary aim of our study was not to establish an accurate measure of aging in mice, but rather to assess the reproducibility of the key findings reported by Johmura et al. (2021) and Wang et al. (2022). These studies suggested that the administration of BPTES (a GLS1 inhibitor) and an anti-PD1 antibody could attenuate aging in elderly mice—findings that have attracted considerable public attention and hold profound societal consequences. Indeed, there are concerns that these agents may soon be adopted as off-label “anti-aging” therapies in private practice.

Given this context, we deliberately focused on reproducing the key datasets presented in those reports. In both studies, p16 expression levels were employed as the key marker of cellular senescence (see Fig. S13A–C in Johmura et al. 2021; Fig. 4a, Extended Data Fig. 7c,d in Wang et al. 2022), while grip strength was adopted as an indicator of organismal aging (see Fig. 4F in Johmura et al. 2021; Fig. 4a, Extended Data Fig. 7f in Wang et al. 2022). [...] Because photographic evidence of rejuvenation phenotypes can strongly influence public perception, we included photographs of all the mice before and after treatment with either BPTES or the anti-PD1 antibody.

We acknowledge, however, that our initial manuscript did not sufficiently explain why we restricted our analysis to p16 expression, grip strength, and the physical appearance of mice. To address this shortcoming, we have now provided a more detailed explanation in the revised manuscript and additionally discuss possible reasons for the discrepancies with previous studies, including inter-individual variation among aged mice and subtle differences in experimental conditions (pp.9-11, lines 187–246). Notably, in the experiments in which aged mice were treated with BPTES, the procedures were performed under the guidance of Dr. Sugimoto, who was directly

responsible for these experiments in Johmura et al. (2021) and is a co-author of the present study. We therefore believe that the methodological discrepancies in this specific experiment were minimized as much as possible.

3. The choice of 18 month old mice might reduce the possible effect size. At 18 month, C57Bl6 are not 'really' old (their typical median lifespan is 25-27 months but ageing rates depend significantly on housing conditions). A possibility could be that the Johmura mice did age faster than the Kawamoto ones, resulting in more discernible effects.

Response:

We appreciate the reviewer's comment. We agree that 80-week-old C57BL/6 mice cannot be considered "truly old." Nevertheless, we chose this age to ensure comparability with Johmura et al. (2021), which employed 80-week-old mice and reported increased p16 expression in liver, lung, and kidney compared to young controls, with reductions following BPTES administration (Fig. S13A–C in Johmura et al. 2021). Although Johmura et al. also examined 24-month-old (~100-week-old) mice, this was restricted to adipose tissue, where abundant macrophages complicated the interpretation due to p16 and SA- β -gal expression independent of senescence (PMID: 28768895). To avoid these confounders, we focused on lung, liver, and kidney in 80-week-old mice. Consistent with Johmura et al., we observed increased p16 expression in these tissues (Fig. 2C in our paper), although we did not detect a reduction after BPTES treatment. While no perfect senescence marker exists, p16 remains the most widely accepted and most directly relevant marker (PMID: 39121846), and both Johmura et al. (2021) and Wang et al. (2022) assessed p16, with Wang et al. relying more heavily on a p16-tdTomato reporter. Thus, we consider our emphasis on p16 appropriate and in line with prior work.

4. Further issues related to the differences between the Johmura and Kawamoto results have in my opinion convincingly dealt with in the response by Johmura et al.

Response:

We appreciate the reviewer's comment. As we have not had access to the response by Johmura et al., we are unable to comment on it directly. Our intention in this study, however, was to provide an independent evaluation of senolytic interventions,

and we believe that such independent replication efforts are valuable for advancing the field.

.

Dear Eiji,

Thank you once more for the submission of your revised manuscript to EMBO Reports. As you know, it has been seen again by the referees who consider their concerns adequately addressed and recommend publication after minor textual revisions.

Before we can proceed with official acceptance, I need you to address the following editorial points:

- Regarding the Author Contributions, we now use CRediT to specify the contributions of each author in the journal submission system. Therefore, please remove the Author Contributions from the manuscript file and make sure that the author contributions in our online manuscript tracking system are correct and up-to-date. The information you specified in the system will be automatically retrieved and typeset into the article. You can enter additional information in the free text box provided, if you wish. See also our guide to authors <https://www.embopress.org/page/journal/14693178/authorguide#authorshipguidelines>.

- Funding information: OU Master Plan Implementation Project under the University of Osaka should be removed from the Comments box and entered as a separate funder via More Funders option in the system.

- Experiments involving mice: please add the reference number(s) for approval (See Author checklist line 95).

- The Data Availability includes the statement: "The source data of this paper are collected in the following database record: " Please add the relevant URL or database record definition.

- The Expanded View figures nomenclature needs correction in the manuscript file and each Figure file: it should be Figure EV1, etc. instead of Expanded View Figure 1, etc.

- Figure 1 and Figure EV1 are low resolution, also in the source data. Please provide high-resolution figures for these two and also higher resolution source data.

- We perform a routine image and data check on all revised manuscript and the following came to our attention:

*Quantification for Figure 1F and EV2C Line 4, Control cells - the numbers for 0nM are repeated for 1uM and 10uM BPTES.

Could you please check whether this is justified? Was the same day control used for all three treatment regimes?

* Quantification for Figure 1G: Day 0/Control for TIG-3 and IMR-90 cells show the same values (4C-4F, 11C-11F). Please check and clarify whether this is correct.

Please see the attached color-coded files.

- Please provide that the exact p values in the legends of figures 2B, C; EV1 A, B, D (unless $p < 0.0001$).

- I read through the manuscript once more and recommend a few changes in the attached document.

- Finally, EMBO Reports papers are accompanied online by

A) a short (1-2 sentences) summary of the findings and their significance,

B) 2-3 bullet points highlighting key results and

C) a schematic summary figure that provides a sketch of the major findings (not a data image).

Please provide the summary figure as a separate file in PNG or JPG format at a size of 550x300-600 pixels (width x height).

Please note that the size is rather small and that text needs to be readable at the final size. Please send us this information along with the revised manuscript.

With kind regards,

Martina

=====

Referee #1:

I think authors addressed my concerns. I recommend publication

Referee #2:

The revisions to the Results and Discussion have appreciably enhanced the scientific clarity of the manuscript, particularly through a more nuanced recognition of the complexity of cellular senescence and the heterogeneity of senescent cell populations in both physiological and pathological settings. The interpretation of the preliminary findings has also been refined in alignment with the reviewers' prior recommendations.

To ensure full consistency across sections, the Abstract should be updated to reflect the revised results and the more balanced conclusions now presented. Additionally, we respectfully request that all occurrences of "n.s." in the figures be replaced with the corresponding exact p-values to improve transparency and statistical rigor.

Referee #3:

My concerns with the first version of the manuscript have been adequately addressed. While I see still relevant limitations (including low numbers of animals, rigor of phenotype characterisation), the study is technically sound and its main thrust is of highest importance for the field.

Thank you for the careful evaluation of our manuscript. We apologize for any inaccuracies or statements that did not fully comply with the journal's policies, and we are grateful for the constructive comments provided by the Editor and the Reviewers. Below, we present our point-by-point responses, with our replies shown in bold.

Editor:

- Regarding the Author Contributions, we now use CRediT to specify the contributions of each author in the journal submission system. Therefore, please remove the Author Contributions from the manuscript file and make sure that the author contributions in our online manuscript tracking system are correct and up-to-date. The information you specified in the system will be automatically retrieved and typeset into the article. You can enter additional information in the free text box provided, if you wish. See also our guide to authors <https://www.embopress.org/page/journal/14693178/authorguide#authorshipguidelines>.

Response:

Thank you for your guidance. We have removed the Author Contributions section from the manuscript file and confirmed that all author contributions in the online submission system are accurate and up-to-date, as requested.

- Funding information: OU Master Plan Implementation Project under the University of Osaka should be removed from the Comments box and entered as a separate funder via More Funders option in the system.

Response:

Thank you for your guidance. We have removed the OU Master Plan Implementation Project under the University of Osaka from the Comments box and entered it as a separate funder via the "More Funders" option in the submission system.

- Experiments involving mice: please add the reference number(s) for approval (See Author checklist line 95).

Response:

We apologize for this omission. We have now added the relevant ethics approval reference numbers to the manuscript as requested.

- The Data Availability includes the statement: "The source data of this paper are collected in the following database record: " Please add the relevant URL or database record definition.

Response:

Thank you for your feedback on the Data Availability statement. We intended this statement to indicate that source data would be made available as supplementary files with the published article. We have not yet deposited the data in an external repository such as BioStudies. We therefore removed the following sentence from the manuscript: "The source data of this paper are collected in the following database record:".

- The Expanded View figures nomenclature needs correction in the manuscript file and each Figure file: it should be Figure EV1, etc. instead of Expanded View Figure 1, etc.

Response:

We have corrected the nomenclature as instructed, changing "Expanded View Figure X" to "Figure EVX" throughout the manuscript and figure files.

- Figure 1 and Figure EV1 are low resolution, also in the source data. Please provide high-resolution figures for these two and also higher resolution source data.

Response:

We have replaced Figure 1 and Figure EV1 with high-resolution versions and have also provided higher-resolution source data for both figures, as requested.

- We perform a routine image and data check on all revised manuscript and the following came to our attention:

*Quantification for Figure 1F and EV2C Line 4, Control cells - the numbers for 0nM

are repeated for 1uM and 10uM BPTES. Could you please check whether this is justified? Was the same day control used for all three treatment regimes?

Response:

Thank you for bringing this to our attention. The identical values were included in error due to an inadvertent copy-and-paste mistake, and this has now been corrected. In addition, to improve transparency, we have revised the source data to report the actual counted cell numbers rather than relative cell numbers (%), and we have included the formula used to calculate the relative values.

* Quantification for Figure 1G: Day 0/Control for TIG-3 and IMR-90 cells show the same values (4C-4F, 11C-11F). Please check and clarify whether this is correct. Please see the attached color-coded files.

Response:

Thank you for drawing our attention to this issue and for providing the color-coded files. We recognize that the identical Day 0 values for TIG-3 and IMR-90 cells in the original dataset could raise concerns during data inspection.

To address this and to avoid any potential ambiguity, we have taken the following steps:

- 1. Figure 1G has been replaced with data from an independent biological replicate, which reproduces the same conclusions while showing clearly distinct baseline values between the two cell lines.**
- 2. All corresponding source data files have been revised to report raw cell counts (actual measured values) instead of relative cell numbers (%), and the calculation formula for the relative cell number (%) has been provided.**

- Please provide that the exact p values in the legends of figures 2B, C; EV1 A, B, D (unless $p < 0.0001$).

Response:

Thank you for this suggestion. Following recent EMBO Reports publications, we have added the exact p values to the figure legends for Figures 2B and 2C, and Figures EV1A, EV1B, and EV1D (except where $p < 0.0001$). Asterisks are retained within the figures for visual clarity. In addition, in response to Reviewer #2's comment, the exact p values corresponding to comparisons previously labeled as "n.s." have also been included in the figure legends.

- I read through the manuscript once more and recommend a few changes in the attached document.

Response:

Thank you for your careful review. We have revised the manuscript in accordance with your comments in the text file.

- Finally, EMBO Reports papers are accompanied online by

A) a short (1-2 sentences) summary of the findings and their significance,

B) 2-3 bullet points highlighting key results and

C) a schematic summary figure that provides a sketch of the major findings (not a data image).

Please provide the summary figure as a separate file in PNG or JPG format at a size of 550x300-600 pixels (width x height). Please note that the size is rather small and that text needs to be readable at the final size. Please send us this information along with the revised manuscript.

Response:

As requested, we have provided (A) a short summary (1–2 sentences) describing the main findings and their significance, (B) 2–3 bullet points highlighting the key results, and (C) a schematic summary figure. The summary and bullet points have been included in the manuscript text, and the schematic summary figure (Graphic Summary) has been submitted as a separate file in PNG format at the requested size.

Referee #1:

I think authors addressed my concerns. I recommend publication

Response:

We sincerely thank the referee for recognizing the significance and central message of our study.

Referee #2:

The revisions to the Results and Discussion have appreciably enhanced the scientific clarity of the manuscript, particularly through a more nuanced recognition of the complexity of cellular senescence and the heterogeneity of senescent cell populations in both physiological and pathological settings. The interpretation of the preliminary findings has also been refined in alignment with the reviewers' prior recommendations.

To ensure full consistency across sections, the Abstract should be updated to reflect the revised results and the more balanced conclusions now presented. Additionally, we respectfully request that all occurrences of "n.s." in the figures be replaced with the corresponding exact p-values to improve transparency and statistical rigor.

Response:

We thank the referee for their thoughtful assessment of our work. As requested, we have revised the final part of the Abstract to better reflect the updated and more balanced conclusions of the manuscript. In addition, we have provided the corresponding exact p values in the figure legends for all instances labeled as “n.s.” in the figures.

Referee #3:

My concerns with the first version of the manuscript have been adequately addressed. While I see still relevant limitations (including low numbers of animals, rigor of phenotype characterisation), the study is technically sound and its main thrust is of highest importance for the field.

Response:

We sincerely thank the referee for recognizing the significance and central message of our study.

Prof. Eiji Hara
The University of Osaka
Institute for Microbial Diseases
3-1, Yamada-oka
Suita
Osaka 565-0871
Japan

Dear Eiji,

Thank you for sending the final revision. I am very pleased to accept your manuscript for publication in the next available issue of EMBO reports. Thank you for your contribution to our journal.

You may qualify for financial assistance for your publication charges - either via a Springer Nature fully open access agreement or an EMBO initiative. Check your eligibility: <https://link.springer.com/journal/44319/how-to-publish-with-us>

Kind regards,

Martina

>>> Please note that it is EMBO Reports policy for the transcript of the editorial process (containing referee reports and your response letter) to be published as an online supplement to each paper. If you do NOT want this, you will need to inform the Editorial Office via email immediately. More information is available here: <https://link.springer.com/partners/embo-press/editorial-policies#Peer%20review>